# PCNA recruits cohesin loader Scc2 to ensure sister chromatid cohesion

Ivan Psakhye ●[1,4] ✉, Ryotaro Kawasumi[1,2,4], Takuya Abe[2], Kouji Hirota ●[2] & Dana Branzei ●[1,3] ✉

Sister chromatid cohesion, established during replication by the ring-shaped multiprotein complex cohesin, is essential for faithful chromosome segregation. Replisome-associated proteins are required to generate cohesion by two independent pathways. One mediates conversion of cohesins bound to unreplicated DNA ahead of replication forks into cohesive entities behind them, while the second promotes cohesin de novo loading onto newly replicated DNA. The latter process depends on the cohesin loader Scc2 (NIPBL in vertebrates) and the alternative PCNA loader CTF18-RFC. However, the mechanism of de novo cohesin loading during replication is unknown. Here we show that PCNA physically recruits the yeast cohesin loader Scc2 via its C-terminal PCNA-interacting protein motif. Binding to PCNA is crucial, as the *scc2-pip* mutant deficient in Scc2–PCNA interaction is defective in cohesion when combined with replisome mutants of the cohesin conversion pathway. Importantly, the role of NIPBL recruitment to PCNA for cohesion generation is conserved in vertebrate cells.

Sister chromatid cohesion (SCC) is mediated by cohesin, one of the three structural maintenance of chromosome (SMC) complexes present in eukaryotic cells[1–4]. SCC is established during replication by cohesin and two genetically defined parallel pathways constituted by replisome components[5–8]. One pathway converts cohesins bound to DNA ahead of the replicative helicases into cohesive complexes, and the other facilitates new cohesin loading onto replicated DNA[9], but the underlying molecular mechanisms are unknown. Similar to other SMC complexes, cohesin is composed of a pair of rod-shaped SMC proteins, Smc1 and Smc3, which heterodimerize via their hinge domains and are connected by a flexible kleisin subunit, Scc1, at their ATPase head domains, thus forming heterotrimeric rings. Structurally, Scc1 serves as a loading platform for three additional essential hook-shaped cohesin subunits called HAWKs (HEAT repeat proteins associated with kleisins): Scc3 interacts with kleisin constitutively, whereas Pds5 and Scc2 (also known as NIPBL) compete for binding[10]. The Scc2 HAWK heterodimerizes with the Scc4 (MAU2 in vertebrates) partner protein, forming a so-called loader complex that promotes cohesin chromatin binding and loop extrusion[11]. In vitro, Scc2 alone can stimulate the ATPase activity of cohesin in the presence of DNA and is sufficient for cohesin DNA loading and loop extrusion[10,12,13]. This involves clamping of DNA by Scc2 and ATP-dependent engagement of cohesin's ATPase head domains[14,15]. Replacement of Scc2 with Pds5 on kleisin abrogates cohesin's ATPase. Our recent results indicate that the essential role of Pds5 in budding yeast is to counteract small ubiquitin-like modifier (SUMO) chain-targeted proteasomal turnover of cohesin on chromatin[16]. Of interest, this essential function of Pds5 is also bypassed by simultaneous loss of the cohesin releaser Wpl1 and the proliferating cell nuclear antigen (PCNA) unloader Elg1 (ref. 16). In attempts to understand the mechanism underlying viability in *elg1Δ wpl1Δ pds5Δ* cells, we discovered that Scc2 possesses a C-terminal PCNA-interacting protein (PIP) motif that recruits the cohesin loader to chromatin during replication to support de novo cohesin loading onto replicated sister DNA and ensure SCC.

## Results

### Scc2 has a C-terminal PCNA-binding motif
We previously showed that kleisin Scc1 is targeted for proteasomal degradation by SUMO chains upon Pds5 loss[16]. Accordingly, fusing

[1]IFOM ETS, the AIRC Institute of Molecular Oncology, Milan, Italy. [2]Department of Chemistry, Graduate School of Science, Tokyo Metropolitan University, Hachioji-shi, Japan. [3]Istituto di Genetica Molecolare, Consiglio Nazionale delle Ricerche, Pavia, Italy. [4]These authors contributed equally: Ivan Psakhye, Ryotaro Kawasumi. ✉e-mail: ivan.psakhye@ifom.eu; dana.branzei@ifom.eu

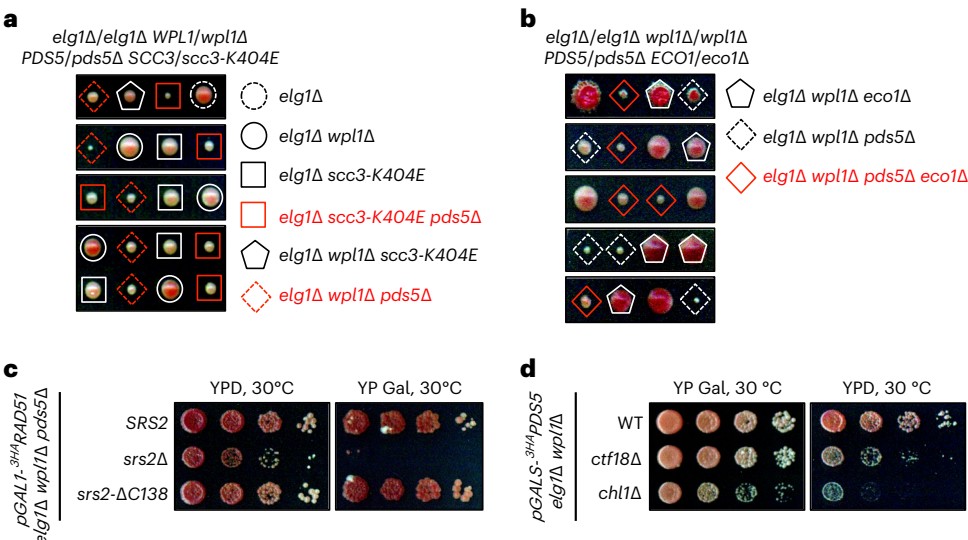

**Fig. 1 | Viability of *elg1Δ wpl1Δ pds5Δ* cells depends on the replisome-associated proteins Ctf18 and Elg1, required to generate cohesion, but not on the Eco1 acetyltransferase. a**, Yeast cells lacking the essential cohesin subunit Pds5 are viable in the absence of both PCNA unloader Elg1 and cohesin releaser Wpl1, or when the latter cannot bind to the Scc3 cohesin subunit due to the Scc3^K404E mutation. **b**, Viability of *elg1Δ wpl1Δ pds5Δ* cells does not depend on Eco1. **c**, Presence of Srs2 helicase, but not its recruitment to PCNA, abolished in the *srs2-ΔC138* mutant, is required to support viability of *elg1Δ wpl1Δ pds5Δ* cells when Rad51 is expressed. **d**, Growth of *elg1Δ wpl1Δ* cells upon *PDS5* shutoff relies on *CTF18* and *CHL1*. WT, wild type.

catalytically active SUMO chain-trimming protease Ulp2 to Scc1 prevents Scc1 turnover and provides viability to cells lacking Pds5 (ref. 16). To address whether specifically Scc1, but not other cohesin subunits, is prone to degradation in the absence of Pds5, we asked whether *GAL* promoter-mediated overexpression of *SCC1* can provide viability in *pds5Δ* mutants. We could retrieve viable *pGAL1-SCC1 elg1Δ pds5Δ* spores upon tetrad dissection but not *pGAL1-SCC1 pds5Δ* double mutants. Moreover, expression of *SCC1* at lower levels due to reduced galactose concentration in the medium resulted in lethality of *pGAL1-SCC1 elg1Δ pds5Δ* cells (Extended Data Fig. 1a). Thus, the essential role of Pds5 is bypassed by loss of the PCNA unloader Elg1 when combined with either loss of the cohesin releaser Wpl1 or with overexpression of the kleisin Scc1, prone to SUMO chain-targeted degradation in *pds5Δ* cells. Interestingly, Wpl1-mediated cohesin unloading requires Pds5 in vitro[17] but *elg1Δ pds5Δ* cells rely on Wpl1 loss for viability[16]. This suggests that Wpl1 can unload cohesin independently of Pds5 in vivo, likely through its interaction with Scc3 (ref. 18). We therefore tested whether Scc3^K404E, defective in binding Wpl1, is also able to provide viability to *elg1Δ pds5Δ* cells, similar to the *wpl1Δ* mutant. This was indeed the case (Fig. 1a). Thus, increasing the chromatin-bound levels of cohesin by preventing its Wpl1-mediated unloading or by overexpressing degradation-prone kleisin Scc1 supports viability of cells lacking Pds5. Notably, these outcomes are only possible when the gene encoding the PCNA unloader Elg1 is additionally deleted.

We then asked how loss of Elg1 contributes to the viability of *elg1Δ wpl1Δ pds5Δ* cells. Increased levels of DNA-loaded PCNA in the absence of Elg1 might recruit the acetyltransferase Eco1 (ref. 19), required to establish SCC by acetylating cohesin[20–24]. By dissecting diploids homozygous for *elg1Δ* and *wpl1Δ* alleles but heterozygous for *ECO1/eco1Δ* and *PDS5/pds5Δ*, we found that Eco1 was not essential for viability in *elg1Δ wpl1Δ pds5Δ* cells (Fig. 1b). The anti-recombinase Srs2 helicase recruited by SUMOylated PCNA was recently suggested to work at replication forks to support the viability of *elg1Δ pds5Δ* cells with increased levels of Scc1 (ref. 25). We indeed observed that *elg1Δ wpl1Δ pds5Δ* cells rely on Srs2 for viability when Rad51 recombinase is expressed. However, this was not the case for the Srs2^ΔC138 mutant proficient in helicase activity but unable to bind PCNA[26] (Fig. 1c and Extended Data Fig. 1b–d). These data suggest that elevated PCNA levels

in the *elg1Δ* background are unlikely to promote SCC through Srs2 recruitment but bring another factor, different from Eco1 (Fig. 1b), to support viability of cells lacking Pds5.

Interestingly, for SCC formation during replication, the presence of the replisome-associated alternative PCNA loader Ctf18-RFC[27,28], which participates in de novo cohesin loading by an unknown mechanism[9], is important[9,29]. We asked whether *elg1Δ wpl1Δ pds5Δ* cells rely on Ctf18 as well as on the Chl1 helicase, a component of the cohesin conversion pathway[9], for viability. Both Ctf18 and Chl1 contributed to normal proliferation following transcriptional shutoff of *PDS5* expressed from the galactose-inducible promoter in *elg1Δ wpl1Δ* cells (Fig. 1d). Because the de novo cohesin loading pathway mediated by Ctf18-RFC requires the cohesin loader Scc2 (ref. 9), we hypothesized that, in *elg1Δ wpl1Δ pds5Δ* cells, the elevated chromatin-bound PCNA pool recruits Scc2 to ensure enough DNA-loaded cohesin for viability. Most PCNA-binding proteins harbor a short sequence motif called PIP that can fit into a cavity on the surface of PCNA[30]. We in fact found a potential PIP motif at the very C terminus of Scc2 (Fig. 2a). Using AlphaFold-Multimer[31], we predicted interaction of the Scc2 PIP with yeast PCNA, similar to the known C-terminal PIP of the DNA polymerase δ nonessential subunit Pol32 (ref. 32) (Fig. 2b). We then mutated the Scc2 PIP with the end result of replacing conserved residues with alanines to generate *scc2-pip*. In vitro glutathione *S*-transferase (GST) pulldown using GST fused with the last 18 amino acids of Scc2 containing the potential PIP revealed that this peptide does indeed interact with PCNA (yeast Pol30), whereas mutated PIP fails to do so (Fig. 2c). Thus, the cohesin loader Scc2 harbors a C-terminal PIP located within the flexible helix on the side opposite to the one with which Scc2 interacts with the Smc1 and Smc3 cohesin subunits (Extended Data Fig. 2a), as judged from the recent cryo-EM structure of the budding yeast cohesin–Scc2–DNA complex[14], in which Scc2 PIP was not resolved due to its mobility. Furthermore, we found that nearly full-length Scc2 fused to GST (amino acids 394–1493, GST–Scc2^C1100), which lacks its largely unstructured N-terminal 393 residues necessary for binding to Scc4, interacts with yeast PCNA in vitro (Extended Data Fig. 2b,c). Notably, this interaction is largely dependent on Scc2 PIP and additionally relies on other residues within the last 168 residues of Scc2 (Fig. 2d), as their truncation abolishes PCNA binding in vitro.

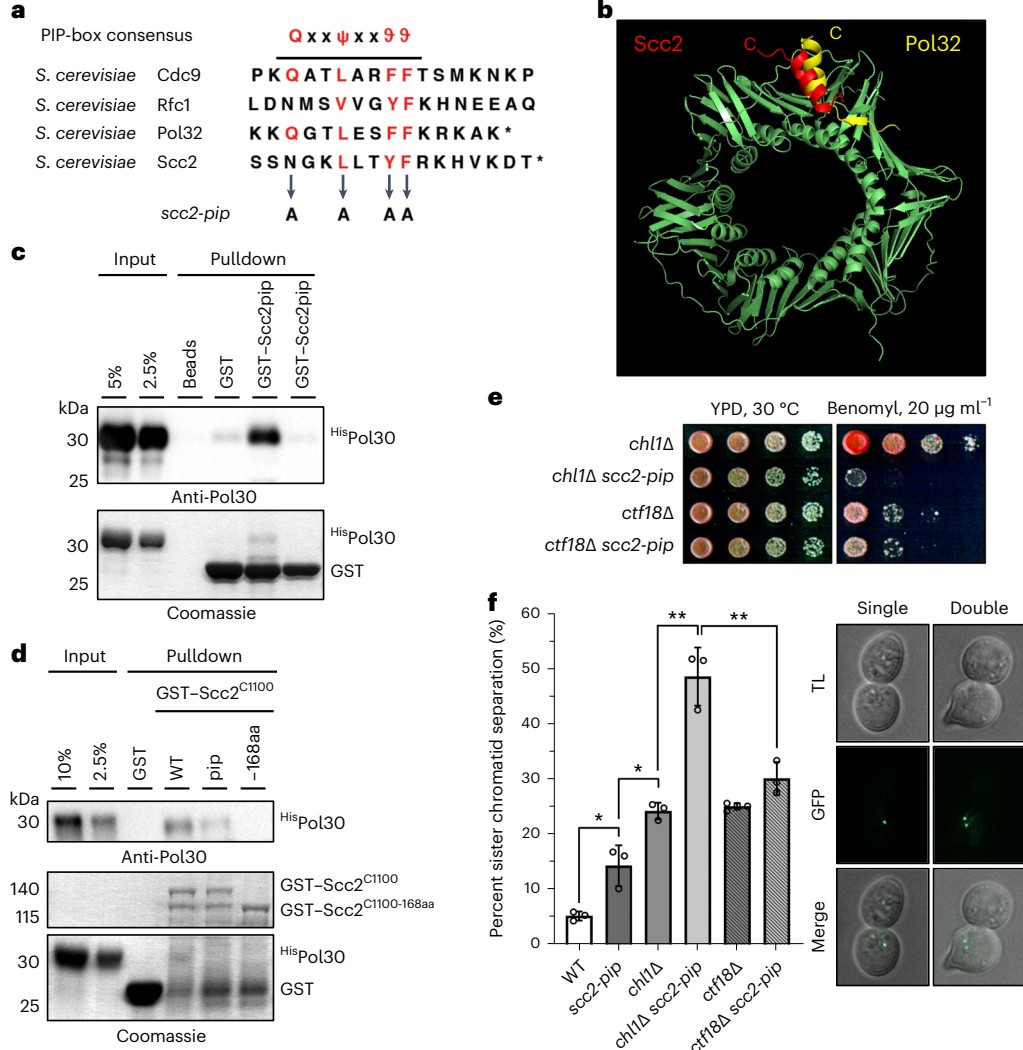

**Fig. 2 | Yeast cohesin loader Scc2 harbors a C-terminal PCNA-binding motif required for SCC. a**, The C terminus of Scc2 contains a consensus PIP motif, where ψ is any hydrophobic residue, ϑ is any aromatic residue, and x is any amino acid. Asterisks indicate the end of the protein sequence. **b**, AlphaFold-Multimer predictions of the interaction between the PCNA homotrimer and the C-terminal PIP of Scc2 (residues 1475–1493; red) and the C-terminal PIP of Pol32 (residues 333–350; yellow). Predictions were aligned using PyMOL, and the C-terminal ends of PIPs are labeled with C. **c**, PIP of Scc2 fused to GST interacts with yeast PCNA ([His]Pol30) in vitro. **d**, In vitro binding of GST–Scc2[C1100] (residues 394–1493) to PCNA largely depends on its PIP and fully relies on residues 1326–1493 of

its C terminus. GST fusion of Scc2 with the last 168 amino acids truncated (GST–Scc2[C1100-168aa]) does not interact with PCNA, whereas mutation of PIP in GST–Scc2[C1100pip] leads to substantial loss of PCNA binding. **e**, The *scc2-pip* mutant shows additive sensitivity to benomyl with *chl1Δ* but is epistatic with *ctf18Δ*. **f**, The *scc2-pip* mutant shows severe cohesion defects in combination with *chl1Δ* but not with *ctf18Δ*. Data are mean values ± s.d. Statistical analysis was performed on results obtained in three independent experiments (*n* = 3) using unpaired two-sided Student's *t*-test; \**P* < 0.014, \*\**P* < 0.0064. At least 240 cells were analyzed for each strain. TL, transmitted light.

## Scc2 PIP acts in cohesin de novo loading

To study the role of Scc2 PIP in SCC, we next generated an *scc2-pip* mutant additionally carrying a C-terminal 6HA tag. Protein from the *scc2-pip* mutant expressed at levels similar with those of wild-type Scc2 (Extended Data Fig. 3a). However, when attempting to obtain *elg1Δ wpl1Δ pds5Δ SCC2-6HA* cells as a control for *elg1Δ wpl1Δ pds5Δ scc2-pip-6HA* cells, we observed that C-terminal tagging alone negatively affects Scc2 function, as deduced from the lethality of the *elg1Δ wpl1Δ chl1Δ SCC2-6HA* mutant (Extended Data Fig. 3b–d). Therefore, we generated the *scc2-pip* mutant without epitope tagging and observed that, similar to the *ctf18Δ* mutant (Fig. 1d), it causes slower proliferation and increased sensitivity to the microtubule poison benomyl in *elg1Δ wpl1Δ pds5Δ* cells (Extended Data Fig. 3e). If Scc2 is recruited via its PIP to PCNA loaded by Ctf18-RFC in the de novo cohesin loading pathway, then one would expect the *scc2-pip* mutant to have an epistatic

relationship with the *ctf18Δ* mutant and negative genetic interactions with mutants of the cohesin conversion pathway (*chl1Δ*, *ctf4Δ*, *csm3Δ*, *tof1Δ*). This was indeed the case when assessed by benomyl sensitivity (Fig. 2e and Extended Data Fig. 3f–h). Moreover, strong additive SCC defects were observed when *scc2-pip* was combined with *chl1Δ* but not with *ctf18Δ* (Fig. 2f), as measured by green fluorescent protein (GFP)-based cytological assays[33]. Thus, the Scc2 PIP has a role in the de novo cohesin loading pathway of SCC.

## Scc2 PIP becomes essential without Scc4

Aside from the interaction with PCNA discovered here, Scc2 localizes robustly to chromatin at centromeres via the binding of its partner Scc4 to the Dbf4-dependent kinase (DDK)-phosphorylated kinetochore protein Ctf19 (ref. 34). To expose the importance of Scc2 PIP for its chromatin localization, we dissected *CTF18*/*ctf18Δ CTF19*/*ctf19Δ* and

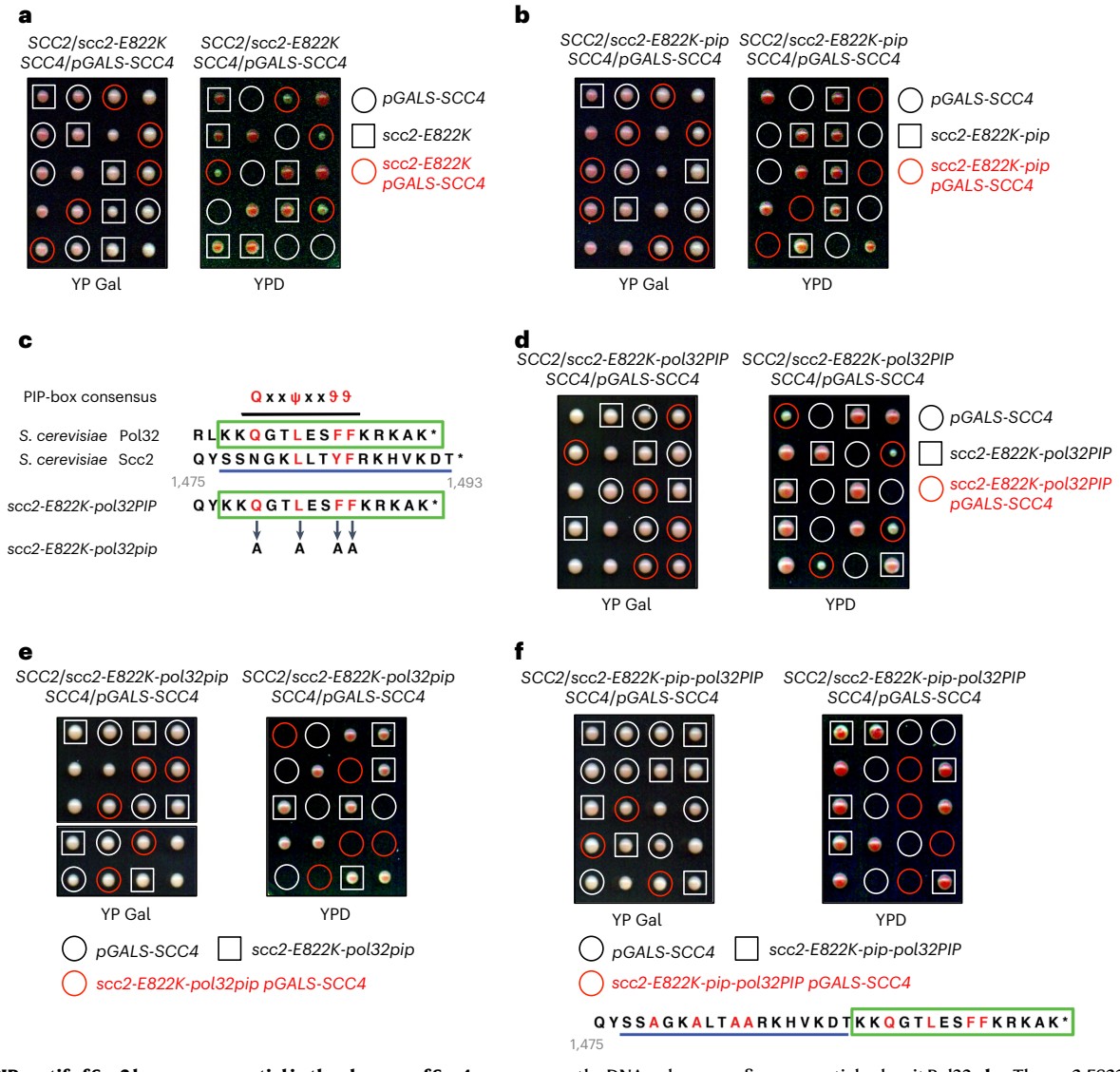

**Fig. 3 | The PIP motif of Scc2 becomes essential in the absence of Scc4.**
**a**, Tetrad-dissection analysis of *SCC2/scc2-E822K SCC4/pGALS-SCC4* diploid cells.
The *scc2-E822K* mutant rescues the lethality of cells expressing *SCC4* from the
galactose-inducible *pGALS* promoter when tetrads are dissected on glucose-
containing yeast extract peptone dextrose (YPD) plates. **b**, The *scc2-E822K-pip*
mutant cannot rescue the lethality of cells upon *SCC4* shutoff. **c**, Scheme
outlining the replacement of endogenous PIP of Scc2 with the C-terminal PIP of

the DNA polymerase δ nonessential subunit Pol32. **d**,**e**, The *scc2-E822K-pol32PIP*
mutant (**d**), but not the *scc2-E822K-pol32pip* mutant having conserved residues
of Pol32 PIP replaced with alanine residues (**e**), rescues the lethality of cells upon
*SCC4* shutoff. **f**, The *scc2-E822K-pip-pol32PIP* mutant carrying a C-terminal fusion
of Pol32 PIP downstream of mutated Scc2 PIP cannot rescue the lethality of cells
upon *SCC4* shutoff.

*CTF18/ctf18Δ CTF19/ctf19Δ scc2-pip/scc2-pip* diploid cells (Extended
Data Fig. 4a,b). Similar to *ctf18Δ*, combining *scc2-pip* with *ctf19Δ*
resulted in additive sensitivity to benomyl (Extended Data Fig. 4c).
Furthermore, *ctf18Δ ctf19Δ scc2-pip* cells grew much more slowly than
*ctf18Δ ctf19Δ* cells but were viable, suggesting that Scc2 is still recruited
to DNA, perhaps through Scc4-mediated binding to the chromatin
remodeler RSC[35]. To completely exclude any possible Scc4-mediated
chromatin localization of Scc2, we decided to use the *scc2-E822K*
mutant that bypasses the requirement of Scc4 for cell viability[10].
Recent work revealed that Scc2 has a key role in clamping DNA onto
engaged SMC heads of cohesin and that Scc2[E822K] might function by
enhancing DNA binding within the clamped state[10,14]. When we dissected
*SCC2/scc2-E822K SCC4/pGALS-SCC4* diploid cells on glucose-containing
plates, *scc2-E822K* suppressed the lethality of Scc4-depleted cells due to
*SCC4* transcriptional shutoff (Fig. 3a). By contrast, the *scc2-E822K-17aa*
mutant lacking the last 17 residues of Scc2 containing PIP (Extended

Data Fig. 4d) and the *scc2-E822K-pip* mutant (Fig. 3b) were no longer
viable upon Scc4 loss. Thus, Scc2 PIP becomes essential in the absence
of Scc4. Importantly, substituting the endogenous Scc2 PIP with the
C-terminal PIP of the DNA polymerase δ nonessential subunit Pol32
(ref. 32) (Fig. 3c) yielded viable Scc4-depleted cells (Fig. 3d) but not
when conserved residues of Pol32 PIP were replaced with alanines
(Fig. 3e). Interestingly, when we fused Pol32 PIP downstream of mutated
endogenous Scc2 PIP, viable *scc4* cells were not produced (Fig. 3f),
further suggesting that other residues of Scc2 might contribute to
interaction with PCNA (Fig. 2d) and that extending the C terminus of
Scc2 precludes the interaction. Finally, instead of mutating Scc2 PIP,
we combined the disassembly-prone PCNA mutant *pol30-D150E*[36,37]
with *pGALS-SCC4 scc2-E822K*. Upon *SCC4* shutoff, cells remained viable
but were highly sensitive to benomyl (Extended Data Fig. 4e), further
supporting the notion that the DNA-bound PCNA pool recruits Scc2
via its PIP to ensure SCC.

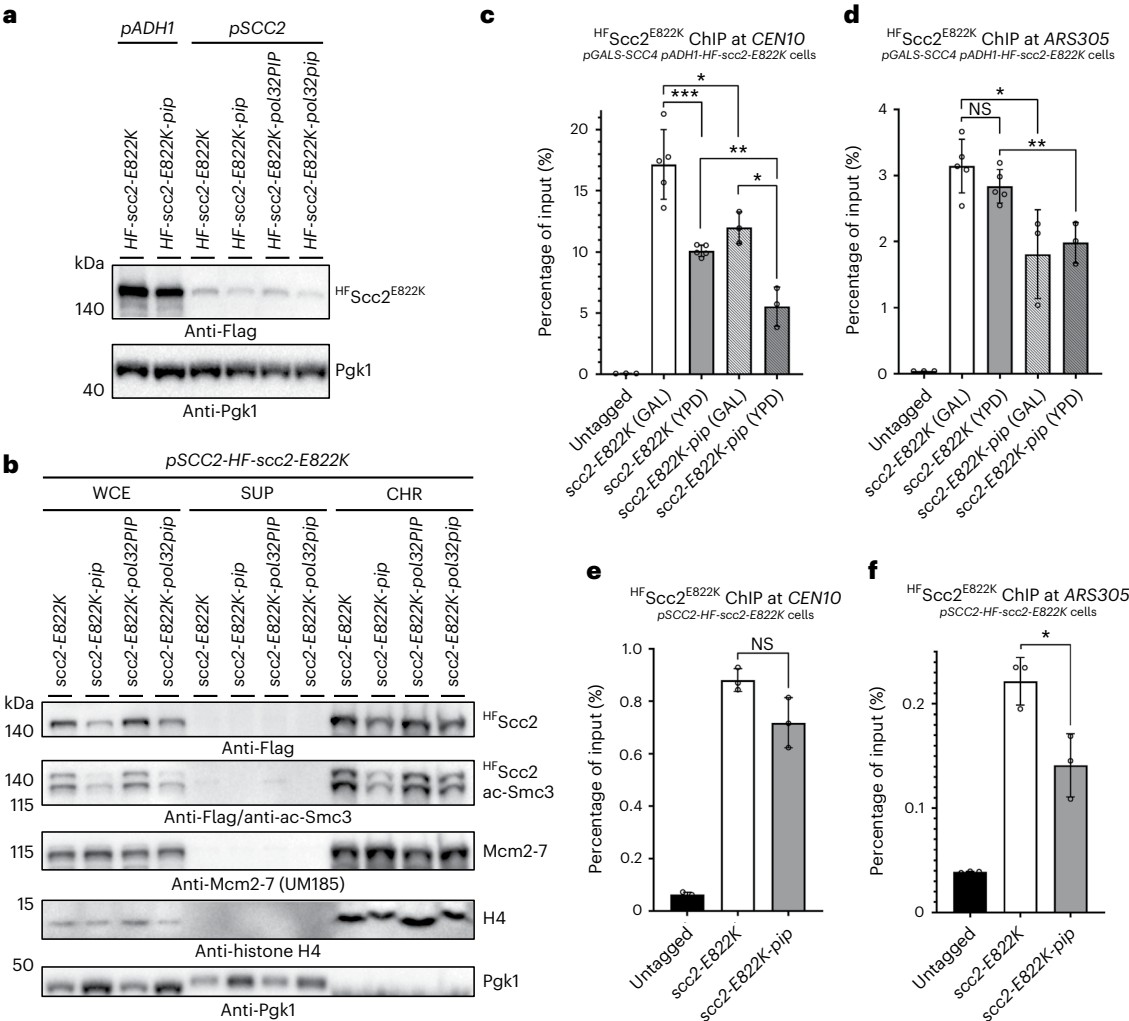

**Fig. 4 | The PIP motif of Scc2 ensures its proper chromatin binding.**
**a**, Protein levels of N-terminally HF-tagged Scc2$^{E822K}$ and its various PIP mutants expressed from either the endogenous promoter *pSCC2* or a strong constitutive promoter, *pADH1*. **b**, Subcellular fractionation of cycling cells expressing HF-tagged Scc2$^{E822K}$ and its various PIP mutants into soluble supernatant (SUP) and chromatin-enriched (CHR) fractions by centrifugation of the whole-cell extract (WCE). Chromatin binding of $^{HF}$Scc2$^{E822K-pip}$ is lower than that of $^{HF}$Scc2$^{E822K}$ and is accompanied by a decrease in acetyl-Smc3 (ac-Smc3) levels. To control chromatin fractionation efficiency, the levels of histone H4, the replicative helicase Mcm2-7 and the cytoplasmic plasma membrane protein Pgk1 were detected in fractions. **c,d**, Loading of $^{HF}$Scc2$^{E822K}$ or its PIP-mutant variant expressed from the strong constitutive promoter *pADH1* onto chromatin at

centromere *CEN10* (**c**) and the centromere-distal early replication origin *ARS305* (**d**) was analyzed by ChIP followed by quantitative PCR (ChIP–qPCR). Used cells had in addition *SCC4* expressed from the galactose-inducible *pGALS* promoter, allowing us to study $^{HF}$Scc2$^{E822K}$ chromatin binding upon *SCC4* shutoff after the shift from galactose-containing medium (GAL) to glucose (YPD). The untagged *scc2-E822K* strain was used as a control. Each ChIP experiment was repeated at least three times (*n* = 3), and each real-time PCR was performed in triplicate. Mean values ± s.d. are plotted. NS, not significant. **e,f**, ChIP–qPCR analysis as in **c,d**, but $^{HF}$Scc2$^{E822K}$ or its PIP-mutant variant as well as *SCC4* are expressed from their endogenous promoters. The mean values of three (*n* = 3) independent experiments ±s.d. are plotted. Statistical analysis was performed using two-sided Student's unpaired *t*-test; *$P < 0.0275$, **$P < 0.006$, ***$P = 0.0006$.

## Scc2 PIP recruits the loader to chromatin

To evaluate the importance of Scc2 PIP for PCNA-guided chromatin recruitment of the cohesin loader, we tagged the N terminus of Scc2$^{E822K}$ with the 7His8Flag (HF) tag and expressed it together with its various PIP mutants from either the strong constitutive promoter *pADH1* or the endogenous *pSCC2* promoter (Fig. 4a). In vitro GST pulldown using GST–PCNA precipitated $^{HF}$Scc2$^{E822K}$ from yeast cell lysates (Extended Data Fig. 5a). Moreover, nickel nitrilotriacetic acid (Ni-NTA) pulldown under denaturing conditions following formaldehyde cross-linking of yeast cultures isolated cross-linked PCNA species in cells expressing $^{HF}$Scc2$^{E822K}$ but not in those expressing $^{HF}$Scc2$^{E822K-pip}$ (Extended Data Fig. 5b). To confirm that the isolated cross-linked species recognized by the Pol30 antibody were indeed PCNA, the C terminus of Pol30 was tagged with 3MYC, and slower-migrating species were detected with

anti-MYC following cross-linking (Extended Data Fig. 5c). To provide further evidence of in vivo interaction between PCNA and Scc2 mediated by the Scc2 PIP, we performed cross-linking using site-specific incorporation of a short photoreactive amino acid[38]. The non-natural photoreactive amino acid *p*-benzoyl-L-phenylalanine (BPA) was incorporated into the C terminus of Scc2, flanking its PIP (Extended Data Fig. 5d), and cross-linked species were detected specifically after ultraviolet light (UV) irradiation of the cells expressing Scc2$^{Q1475BPA}$ or Scc2$^{T1493BPA}$ (Extended Data Fig. 5e). Interestingly, combining *POL30-3MYC* with *scc4Δ scc2-E822K* resulted in increased benomyl sensitivity (Extended Data Fig. 6a), suggesting that tagging PCNA might affect Scc2 binding. Indeed, using AlphaFold-Multimer[31], we predicted that tagging the C terminus of PCNA with 3MYC interferes with its binding to Scc2 PIP (Extended Data Fig. 6b) by affecting the so-called front face

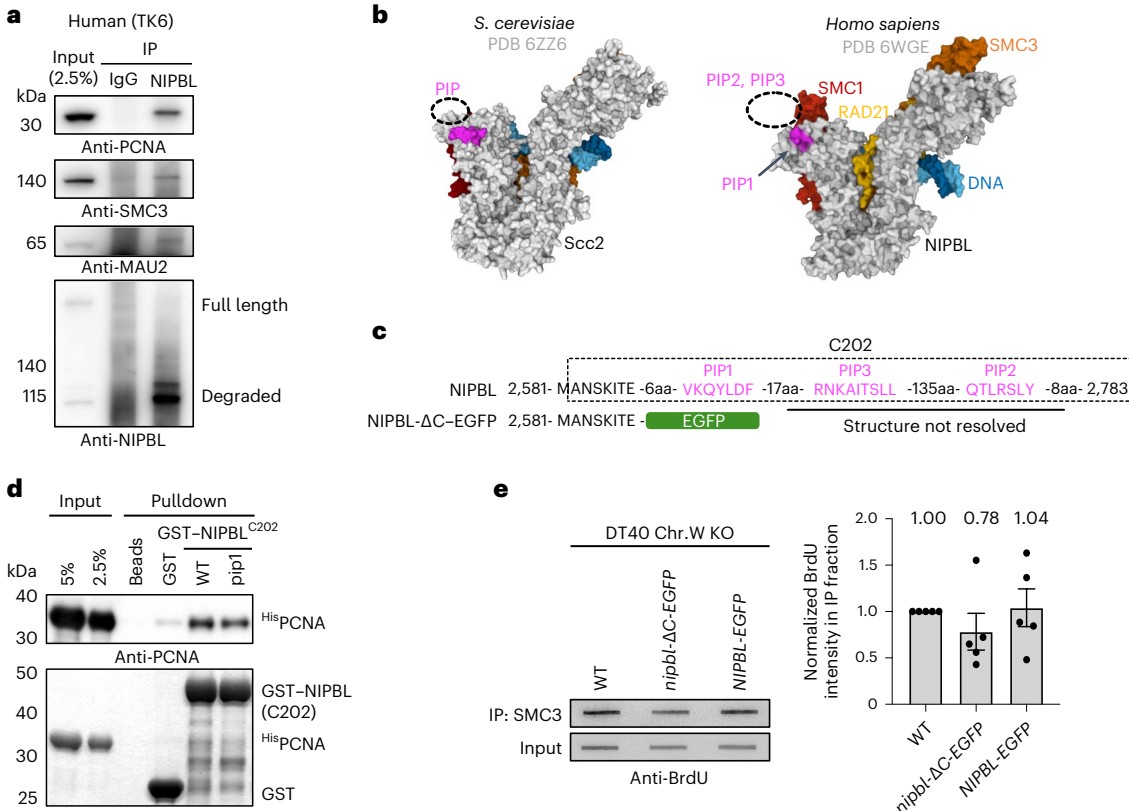

**Fig. 5 | PCNA-mediated recruitment to chromatin is conserved in the human and chicken cohesin loader NIPBL. a**, Immunoprecipitation (IP) of human NIPBL in TK6 cells. PCNA was co-immunoprecipitated with NIPBL. SMC3 and MAU2 are shown as positive controls. **b**, Cryo-EM structures of budding yeast Scc2–cohesin (PDB 6ZZ6) and human NIPBL–cohesin (PDB 6WGE). DNA is colored blue; SMC1, red; SMC3, orange; RAD21, yellow. Ten amino acids (FSAQLENIEQ) upstream of budding yeast Scc2 PIP, unresolved in cryo-EM structure due to its mobility, and PIP1 of human NIPBL are colored pink. Dashed circles indicate where unresolved PIPs should be positioned. **c**, The C terminus

of chicken NIPBL containing three PIP-like motifs. *nipbl-ΔC-EGFP* cells lack the last 195 amino acids of NIPBL, and the EGFP tag is fused instead. aa, amino acids. **d**, A C-terminal fragment of NIPBL (C202) fused to GST interacts with chicken PCNA in vitro. Mutating PIP1 weakens the interaction. **e**, The amount of SMC3 on newly replicated chromatin was determined by the BrdU–ChIP–slot–western technique. Truncation of the last 195 residues of NIPBL results in a reduction of SMC3 levels bound to nascent DNA. The mean values of five independent experiments ±s.e.m. are plotted. Chr.W, chromosome W; KO, knockout.

of the PCNA clamp. This prompted us to check the mutants *pol30-6* and *pol30-79* that cause disruptions of a surface cavity on the front face of the PCNA ring[39], which might contribute to Scc2–PCNA interaction in addition to Scc2 PIP. Combining *pol30-6* and *pol30-79* with *scc4Δ scc2-E822K* resulted in synthetic sickness and increased benomyl sensitivity (Extended Data Fig. 6c–e). In summary, these results confirm the physical interaction between PCNA and Scc2 mediated by the front face of the PCNA ring and Scc2 PIP.

Next, we assayed the chromatin-binding properties of [HF]Scc2[E822K] and its various PIP mutants expressed from the endogenous promoter *pSCC2* using chromatin fractionation (Fig. 4b), after having confirmed that N-terminal tagging does not change their genetic interaction with *scc4* (Extended Data Fig. 7a–d) compared to untagged *scc2* mutants (Fig. 3). We observed reduced chromatin binding of [HF]Scc2[E822K-pip] compared to that of [HF]Scc2[E822K]. This decrease in chromatin binding was accompanied by decreased levels of chromatin-bound acetylated Smc3, indicative of the overall cohesive pool of cohesin in cells. Substitution of endogenous Scc2 PIP with Pol32 PIP supported normal chromatin binding and acetylated Smc3 levels, which was not the case for mutated Pol32 PIP (Fig. 4b). Moreover, *SCC4* shutoff caused a reduction in chromatin binding of [HF]Scc2[E822K], which was additive with the decrease caused by the *scc2-pip* mutation (Extended Data Fig. 7e). Thus, both Scc4 and Scc2 PIP contribute to the chromatin binding of the cohesin loader. Finally, we quantitatively assayed [HF]Scc2[E822K]

binding at the centromere *CEN10* and the early origin of replication *ARS305* by chromatin immunoprecipitation (ChIP). Upon *SCC4* shutoff (change from galactose-containing medium to YPD), the binding of [HF]Scc2[E822K] and its PIP mutant expressed from the strong *pADH1* promoter dropped specifically at the centromere but not at the replication origin (Fig. 4c,d). Mutation of Scc2 PIP reduced its binding at both locations. When [HF]Scc2[E822K] and its PIP mutant were expressed from the endogenous *pSCC2* promoter, their ChIP efficiency dropped tenfold at both genomic loci (Fig. 4e,f), and [HF]Scc2[E822K-pip] showed a statistically significant reduction in chromatin binding at *ARS305* but not at *CEN10*. Thus, Scc2 expression levels affect the overall degree of its chromatin binding. Importantly, the results indicate that Scc2 PIP contributes to cohesin loader recruitment to replication origins, whereas Scc4 is more important for Scc2 localization at centromeres.

### PCNA-guided Scc2 recruitment is conserved

We next asked whether the same mechanism is conserved in vertebrate cells. To this end, we performed co-immunoprecipitation experiments, revealing that NIPBL indeed interacts with PCNA in human TK6 cells as well as in chicken DT40 cells (Fig. 5a and Extended Data Fig. 8a). Moreover, cell cycle synchronization followed by co-immunoprecipitation showed more binding of NIPBL to PCNA in S and G2/M phase than in asynchronous or G1 phase cell populations (Extended Data Fig. 8b). Based on the cryo-EM structures of

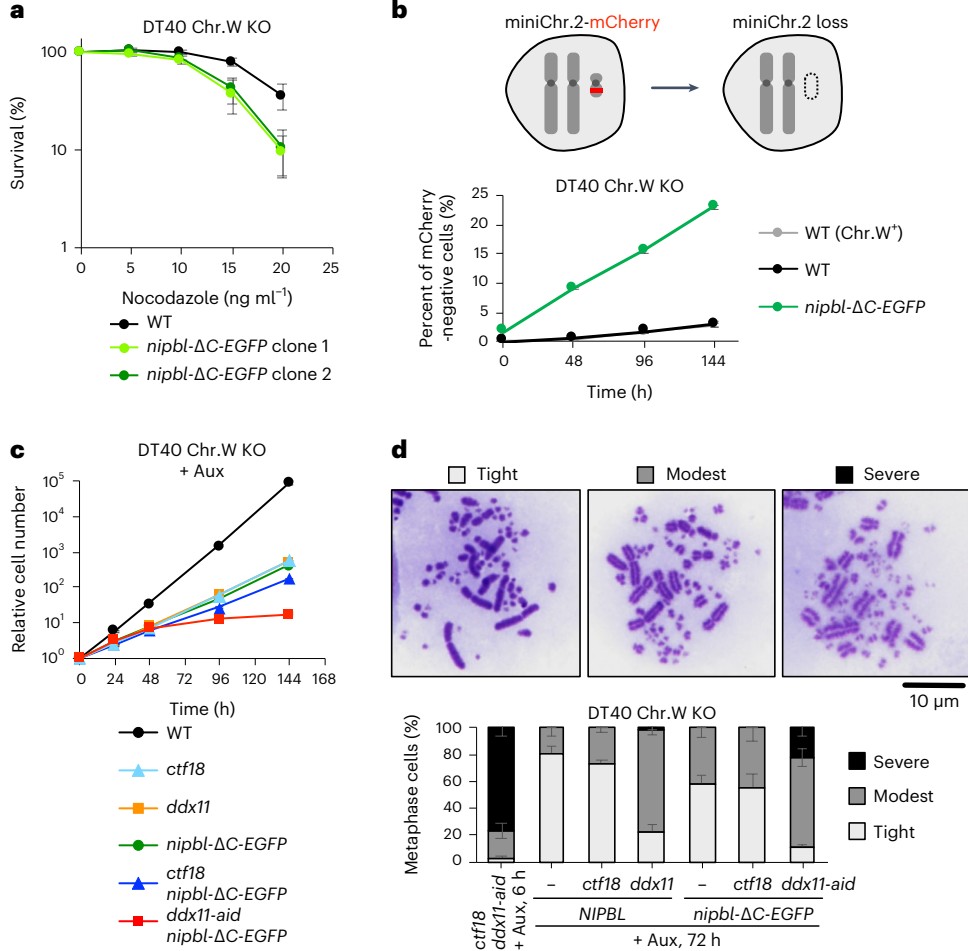

**Fig. 6 | PCNA-guided recruitment of the chicken cohesin loader NIPBL to chromatin is required for SCC. a**, Sensitivity assay using CellTiter-Glo. The *nipbl-ΔC-EGFP* mutant exhibits hypersensitivity to nocodazole treatment. The mean values of three independent experiments ±s.d. are plotted. **b**, Frequency of chromosome loss was measured by the minichromosome-loss assay. Minichromosome 2 (miniChr.2) carries an mCherry expression unit, and the percentage of mCherry-negative cells was determined by flow cytometry. The mean values of three independent experiments ±s.d. are plotted. **c**, Growth curves showing synthetic lethality of *nipbl-ΔC-EGFP* with *ddx11-aid* but not with *ctf18*. The mean values of three independent experiments ±s.d. are plotted. Aux, auxin. **d**, Cohesion analysis of metaphase spreads. Depletion of DDX11 in the *nipbl-ΔC-EGFP* background results in severe cohesion defects. The mean values of three independent experiments ±s.d. are plotted.

Scc2 or NIPBL, structure predictions and motif analysis[14,15,40-42], we found three potential PIP-like motifs at the NIPBL C terminus resembling the Scc2 PIP (Fig. 5b,c and Extended Data Fig. 8c,d). In vitro GST pulldown assays using recombinant chicken PCNA and the NIPBL C-terminal fragment (C202) confirmed the direct interaction (Fig. 5c,d). Furthermore, fusion of full-length NIPBL (residues 1–2783) to GST showed binding to chicken PCNA in vitro, whereas truncation of the last 195 amino acids of NIPBL, containing PIP-like motifs, (residues 1–2588; GST–nipbl-ΔC) reduced the interaction (Extended Data Fig. 8e), similar to *scc2-pip* (Fig. 2d). Because NIPBL PIP1 more closely resembles the yeast Scc2 PIP among the three PIP-like motifs, based on its location, we mutated PIP1 and tested the effect on its interaction with PCNA. The interaction was only moderately reduced (Fig. 5d), implying the contribution of other potential PIP-like motifs. In fact, all PIP-like motifs interacted with PCNA in vitro, and the interaction was weakened when they were mutated (Extended Data Fig. 8f). Furthermore, using AlphaFold-Multimer[31], we predicted binding of NIPBL PIP2 and PIP3 to chicken PCNA, similar to PCNA interacting with the flexible C-terminal PIP in the budding yeast cohesin loader Scc2 and the fission yeast cohesin loader Mis4 (Extended Data Fig. 9), not resolved in cryo-EM structures previously[14,15,43] due to their mobility.

Next, to assess the consequences of the defect in PCNA–NIPBL interaction, we established *nipbl-ΔC-EGFP* cells, in which the last 195 residues of NIPBL are replaced with EGFP (Fig. 5c). Because chicken *NIPBL* genes are located on chromosomes Z and W, we used DT40 cells lacking chromosome W, which thus carry a single copy of *NIPBL* (Extended Data Fig. 10a). Subsequently, the EGFP tag was introduced to the C terminus of NIPBL using the Flp-In system, with or without truncating the last 195 residues where the three PIP-like motifs are positioned (Extended Data Fig. 10b–d). We then employed the 5-bromodeoxyuridine (BrdU)–ChIP–slot–western technique[44] to monitor cohesin recruitment to newly replicated chromatin. Notably, the amount of BrdU co-immunoprecipitated with SMC3 was reduced in *nipbl-ΔC-EGFP* cells, suggesting that the PCNA–NIPBL interaction facilitates cohesin loading on nascent chromatin (Fig. 5e). Moreover, *nipbl-ΔC-EGFP* cells exhibited hypersensitivity to nocodazole (Fig. 6a and Extended Data Fig. 10e) and increased chromosome loss rate (Fig. 6b), measured by the minichromosome-loss assay[45]. Thus, these results confirm the importance of PCNA-guided NIPBL recruitment for proper SCC.

The *scc2-pip* mutant and mutants of the de novo cohesin loading pathway factors showed epistatic genetic interaction in budding yeast

(Fig. 2e,f). To test whether the same holds true in vertebrates, we aimed to knock out *CTF18* and *DDX11* in the *nipbl-ΔC-EGFP* mutant. Notably, we could disrupt *CTF18* but not *DDX11* (Extended Data Fig. 10f). Therefore, we employed the auxin-inducible degron (AID) system to obtain *ddx11-aid* conditional mutants[46,47] (Extended Data Fig. 10g). Remarkably, *ddx11-aid nipbl-ΔC-EGFP* cells stopped proliferating 4 d after auxin addition (Fig. 6c), indicative of synthetic lethality. Moreover, strong additive SCC defects were observed when *nipbl-ΔC-EGFP* was combined with DEAD/H-box helicase 11 (DDX11) depletion but not with *CTF18* knockout (Fig. 6d), consistent with our findings in yeast (Fig. 2f). In summary, we conclude that PCNA-guided recruitment of Scc2 (NIPBL in vertebrates) onto replicated DNA is a fundamental mechanism to ensure SCC, conserved from yeast to vertebrates.

## Discussion

Our study uncovers a conserved mechanism of de novo cohesin loading during DNA replication necessary for SCC. This mechanism relies on the recruitment of the cohesin loader Scc2 (or NIPBL) to the PCNA sliding clamp, deposited onto newly replicated DNA by the alternative PCNA loader CTF18-RFC associated with the replisome. Previously, cohesin and NIPBL loaded at replication origins were proposed to remain associated with the replicative helicase MCM and then transferred behind the replication fork to establish SCC[48]. Our work suggests that PCNA, the maestro of various replication-linked functions, serves as a recruiting platform for incoming Scc2. Thus, PCNA coordinates Scc2-mediated cohesin de novo loading onto replicated DNA, with cohesin acetylation required for SCC establishment and mediated by Eco1 (ESCO2 in vertebrates) acetyltransferase[20–24], likewise recruited by the homotrimeric PCNA ring[49,50].

## Online content

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

## Methods

### Yeast strains, techniques and growing conditions

Chromosomally tagged *Saccharomyces cerevisiae* strains and mutants were constructed using a PCR-based strategy, with genetic crosses and standard techniques[51]. Standard cloning and site-directed mutagenesis techniques were used. Strains and all genetic manipulations were verified by PCR, sequencing and phenotype. Maps and primer DNA sequences are available upon request. All yeast strains used in this work are isogenic to the W303 background and are listed in the Supplementary Table 1. Yeast cultures were inoculated from overnight cultures and grown using standard growth conditions and media[52]. All cultures were grown in YPD medium containing glucose (2%) as a carbon source at 30 °C unless otherwise indicated. For transcriptional shutoff of genes expressed under the control of the *GAL* promoter, cells were grown in YP Gal medium containing galactose (2%), washed once with 1× PBS and transferred to YPD medium or plated on YPD plates. For cell cycle synchronization, logarithmic cells grown at 30 °C were arrested in G1 phase using 3–5 µg ml$^{-1}$ α-factor for 2–3 h. G2/M arrest was performed with 20 µg ml$^{-1}$ nocodazole for 2–3 h. G1 or G2/M arrest was verified microscopically and by flow cytometry analysis. For drug-sensitivity assays, cells from overnight cultures were counted and diluted before being spotted on YPD plates containing the indicated concentrations of benomyl and incubated at 30 °C for 2–3 d. The tetrad-dissection analysis was performed using the Singer Instruments MSM 400 system on YPD or YP Gal plates.

### Premature sister chromatid-separation assay in budding yeast

SCC was measured as described previously[33]. Logarithmically growing cells were treated with 3 µg ml$^{-1}$ α-factor to induce G1 arrest. Cells were then washed with YP medium and released in YPD containing 20 µg ml$^{-1}$ nocodazole to allow one round of replication. After 3 h of nocodazole treatment, G2/M arrest was checked by cell morphology, and cells were collected, washed once with 1× PBS and fixed in 70% ethanol overnight at −20 °C. Cells were then resuspended in 50 mM Tris-HCl, pH 6.8 and sonicated for 5 s before microscopic analysis. Cells were imaged on a DeltaVision microscope (Applied Precision) using a ×100 oil-immersion lens. Images were analyzed using ImageJ software. Statistical analysis was performed on results obtained in at least three independent experiments using two-sided Student's unpaired *t*-test, in which at least 240 cells were analyzed for each strain. The error bars represent s.d.

### GST in vitro pulldown assays

The sequence for the C terminus of yeast Scc2 (residues 1476–1493), either wild type (termed GST–Scc2PIP) or with PIP mutated (Scc2$^{N1479A,L1482A,Y1485A,F1486A}$, termed GST–Scc2pip), was introduced into the pGEX-6P-1 (GE Healthcare) vector, replacing the sequence for the last 18 residues of GST. The sequence for nearly full-length Scc2 (amino acids 394–1493), which lacks its largely unstructured N-terminal 393 residues necessary for binding to Scc4, was fused to the sequence for GST (termed GST–Scc2$^{C1100}$) in pGEX-6P-1, and Scc2 PIP was either mutated (GST–Scc2$^{C1100pip}$), or the last 168 amino acids of Scc2 were truncated (GST–Scc2$^{C1100-168aa}$). The sequence for full-length chicken NIPBL (residues 1–2783) was fused to the sequence for GST (termed GST–NIPBL) in pGEX-6P-1. The C-terminal truncation of NIPBL (residues 1–2588) that lacks the last 195 amino acids containing PIP-like motifs, fused to GST, was termed GST–nipbl-ΔC. The sequence for the C terminus of chicken NIPBL (residues 2582–2783), either wild type (termed GST–NIPBL$^{C202WT}$) or with PIP1 mutated (NIPBL$^{Q2597A,L2599A,F2601A}$, termed GST–NIPBL$^{C202pip1}$), was introduced into the pGEX-6P-1 vector, replacing the sequence for the last 13 residues of GST. In addition, sequences containing individual potential PIP motifs of the chicken NIPBL C terminus (amino acids 2586–2608, termed GST–NIPBL-PIP1; amino acids 2764–2783, termed GST–NIPBL-PIP2; amino acids 2616–2635, termed GST–NIPBL-PIP3) or having sequences for their conserved residues mutated (NIPBL$^{Q2597A,L2599A,F2601A}$, termed GST–NIPBL-pip1; NIPBL$^{Q2769A,L2771A,Y2775A}$,

termed GST–NIPBL-pip2; NIPBL$^{R2626A,N2627A,I2630A,L2633A,L2634A}$, termed GST–NIPBL-pip3) were introduced into the pGEX-6P-1 vector, replacing the sequence for the last 13 residues of GST. Rosetta (DE3) pLysS competent *Escherichia coli* cells (Novagen) were used for protein expression. Following overnight protein induction with 0.25 mM IPTG at 16 °C in 100-ml cell cultures, cells were pelleted, resuspended in 6 ml lysis buffer (1× PBS, 500 mM NaCl, 1% Triton X-100, lysozyme, Calbiochem EDTA-free Protease Inhibitor Cocktail Set III) and sonicated on ice. The crude lysate was clarified by centrifugation at 21,000*g* for 15 min at 4 °C, and the supernatant was mixed with 0.2 ml of glutathione Sepharose 4B beads (GE Healthcare) pre-equilibrated with lysis buffer. Following overnight incubation at 4 °C, five washes with lysis buffer were performed, and the beads with bound GST fusion proteins were used for subsequent in vitro pulldown assays with recombinant PCNA (either yeast N-terminally His-tagged Pol30 with yeast Scc2GST fusions or chicken N-terminally His-tagged PCNA with chicken NIPBL GST fusions). The amounts of GST fusion proteins bound to the beads were estimated by comparison to BSA samples of known concentrations resolved by SDS–PAGE and Coomassie blue staining. To study the interaction between GST–Scc2PIP fusions and PCNA, purified recombinant His-tagged yeast Pol30 (2.5 µg) was incubated either with GST–Scc2PIP (2.5 µg) and its PIP-mutant variant GST–Scc2pip or GST (2.5 µg) alone bound to glutathione Sepharose 4B beads in 0.7 ml binding buffer (1× PBS, 150 mM NaCl, 1% Triton X-100, Calbiochem EDTA-free Protease Inhibitor Cocktail Set III) overnight at 4 °C, with gentle mixing. After the incubation, beads were washed five times with 1 ml binding buffer, and bound proteins were eluted with 50 µl HU sample buffer. Samples were then analyzed by SDS–PAGE, followed by western blotting and probing with anti-Pol30 (GTX64144, GeneTex) and, in parallel, by staining the protein gel with Coomassie blue. To study the interaction between full-length Scc2 and PCNA, the recombinant GST–Pol30 fusion was purified and used for GST pulldown from whole-cell extracts obtained by grinding yeast cells in liquid nitrogen expressing N-terminally HF-tagged Scc2$^{E822K}$ ($^{HF}$Scc2$^{E822K}$) from a strong constitutive *pADH1* promoter. Anti-PCNA (sc-25280, Santa Cruz Biotechnology) and anti-GST were used to study the interaction between chicken PCNA and GST–NIPBL fusions.

### Trichloroacetic acid protein precipitation

For preparation of denatured protein extracts, yeast cultures grown to an optical density at 600 nm (OD$_{600}$) of 0.7–1 were pelleted by centrifugation (1,500*g*, 4 min, 4 °C) and immediately frozen in liquid nitrogen. After thawing on ice, the pellets were lysed by addition of denaturing lysis buffer (1.85 M NaOH, 7.5% β-mercaptoethanol) for 15 min on ice. For a cell pellet from 1 ml culture with an OD$_{600}$ of 1, typically 150 µl lysis buffer was used. To precipitate proteins, the lysate was subsequently mixed with an equal volume (150 µl for 1 ml culture with an OD$_{600}$ = 1) of 55% (wt/vol) trichloroacetic acid (TCA) and further incubated on ice for 15 min. The precipitated material was recovered by two sequential centrifugation steps (16,000*g*, 4 °C, 15 min). Pelleted denatured proteins were then directly resuspended in HU sample buffer (8 M urea, 5% SDS, 1 mM EDTA, 1.5% DTT, 1.5% bromophenol blue; 50 µl for cell pellet from 1 ml culture with an OD$_{600}$ = 1), boiled for 10 min and stored at −20 °C. Proteins were resolved on precast Bolt 4–12% Bis-Tris Plus gradient gels and analyzed by standard western blotting techniques. Bio-Rad Image Lab version 5.2.1 was used for western blot acquisition. Mouse monoclonal anti-Flag (1:2,000, clone M2) was purchased from Sigma-Aldrich. Mouse monoclonal anti-Pgk1 antibody (1:2,000, clone 22C5D8) was obtained from Thermo Fisher Scientific. Mouse monoclonal anti-HA (1:2,000, clone F-7) and anti-PCNA (1:2,000, clone F-2) were from Santa Cruz Biotechnology as well as normal mouse IgG. Rabbit polyclonal anti-Pol30 antibody (1:2,000, GTX64144) was purchased from GeneTex. Mouse monoclonal anti-c-MYC (1:2,000, clone 9E10) and rabbit polyclonal anti-GST (1:2,000) were produced in house. Rabbit polyclonal anti-histone H4 (1:2,000, ab7311) was obtained from

Abcam. Mouse monoclonal anti-acetyl-Smc3[53] (1:2,000) was a gift from K. Shirahige. Rabbit polyclonal anti-Mcm2-7[54] (1:5,000, UM185) was a gift from S. P. Bell. Anti-rabbit IgG and anti-mouse IgG, HRP-linked antibodies (1:5,000), were purchased from Cell Signaling Technology.

## Ni-NTA pulldown of HF-tagged Scc2^E822K after formaldehyde cross-linking of yeast cells

For isolation of Scc2 protein interactors from yeast cells expressing N-terminally HF-tagged Scc2^E822K ($^{HF}$Scc2^E822K) or its PIP-mutant variant, denatured protein extracts were prepared following formaldehyde cross-linking, and Ni-NTA chromatography was carried out as described previously[55,56]. Briefly, 200 ml yeast cell culture with an $OD_{600} = 1$ of logarithmically growing cells was collected by centrifugation (1,500$g$, 4 min, 4 °C) after 30 min of cross-linking with 1% formaldehyde, washed with pre-chilled water, transferred to a 50-ml Falcon tube and lysed with 6 ml of 1.85 M NaOH, 7.5% β-mercaptoethanol for 15 min on ice. Proteins were precipitated by adding 6 ml of 55% TCA and incubating for another 15 min on ice (TCA precipitation, described above). Next, the precipitate was pelleted by centrifugation (1,500$g$, 15 min, 4 °C), washed twice with water and finally resuspended in buffer A (6 M guanidine hydrochloride, 100 mM NaH$_2$PO$_4$, 10 mM Tris-HCl, pH 8.0, 20 mM imidazole) containing 0.05% Tween-20. After incubation for 1 h on a roller at room temperature with subsequent removal of insoluble aggregates by centrifugation (23,000$g$, 20 min, 4 °C), the protein solution was incubated overnight at 4 °C with 50 μl Ni-NTA agarose beads in the presence of 20 mM imidazole. After incubation, the beads were washed three times with buffer A containing 0.05% Tween-20 and five times with buffer B (8 M urea, 100 mM NaH$_2$PO$_4$, 10 mM Tris-HCl, pH 6.3) containing 0.05% Tween-20. $^{HF}$Scc2^E822K and its cross-linked species bound to the beads were finally eluted by incubation with 50 μl HU sample buffer for 10 min at 65 °C. Proteins were resolved on precast Bolt 4–12% Bis-Tris Plus gradient gels and analyzed by standard western blotting techniques.

## In vivo photo-cross-linking followed by protein immunoprecipitation

In vivo BPA cross-linking has been successfully used to identify interaction sites of different subunits of the cohesin complex[14] and was carried out as described previously[38]. Briefly, an amber stop codon (TAG) was incorporated into specific sites of the sequence for N-terminally 3HA-tagged Scc2 flanking its C-terminal PIP, thus replacing the natural residues at desired positions (Q1475 or T1493). Yeast stains expressing C-terminally 3MYC-tagged Pol30 (yeast PCNA) and TAG-substituted 3HA–Scc2 variants and additionally carrying plasmid pLH157 (*TRP1 EcTyrRS EctRNACUA*) were grown in −Trp medium containing 0.25 mM BPA (Bachem) as follows. A liquid culture was grown for 12 h in −Trp medium with BPA. Cells were diluted to an $OD_{600}$ of 0.05 in 300 ml of −Trp medium with BPA and grown overnight to an $OD_{600}$ of 0.7. Cells were then collected by centrifugation at 1,500$g$ for 5 min at 4 °C, pellet of 100 ml cell culture at an $OD_{600} = 1$ was frozen in liquid nitrogen, and the remaining pellet of 100 ml cell culture at an $OD_{600} = 1$ was resuspended in 15 ml of ice-cold PBS buffer. The cell suspension was transferred into three wells of a six-well tissue culture plate (Falcon), placed on ice in the UV Stratalinker 2400 Crosslinker (Stratagene) and irradiated at 365 nm (3×, 300 s of UV followed by a 5-min rest on ice and resuspension). After irradiation, cells were collected by centrifugation and frozen in liquid nitrogen. For the immunoprecipitation, yeast protein extracts were prepared by cell disruption using grinding in liquid nitrogen. To avoid protein degradation, lysis buffer (150 mM NaCl, 10% glycerol, 1% NP-40, 50 mM Tris-HCl, pH 8.0) was supplemented with inhibitors: EDTA-free complete cocktail, 20 mM $N$-ethylmaleimide, 1 mM phenylmethanesulfonyl fluoride and 25 mM iodoacetamide. For immunoprecipitations, anti-HA and anti-MYC together with recombinant protein G Sepharose 4B beads were used. Immunoprecipitations were performed overnight with head-over-tail rotation at 4 °C and were followed by stringent washing steps to remove nonspecific background binding to the beads.

## Chromatin immunoprecipitation followed by quantitative PCR

ChIP was carried out as previously described[57]. Briefly, cells were collected under the indicated experimental conditions and cross-linked with 1% formaldehyde for 30 min. Cells were washed twice with ice-cold 1× TBS, suspended in lysis buffer supplemented with 1 mM phenylmethyl sulfonyl fluoride (PMSF), 20 mM NEM and 1× EDTA-free complete cocktail and lysed using the FastPrep-24 system (MP Biomedicals). Chromatin was sheared to a size of 300–500 bp by sonication. Immunoprecipitation reactions with anti-Flag and Dynabeads protein G were allowed to proceed overnight at 4 °C. After washing and eluting the ChIP fractions from beads, cross-links were reversed at 65 °C overnight for both input and immunoprecipitated samples. After proteinase K treatment, DNA was extracted twice with phenol–chloroform–isoamyl alcohol (25:24:1, vol/vol). Following precipitation with ethanol and ribonuclease A (RNase A) treatment, DNA was purified using the QIAquick PCR Purification Kit. Real-time PCR was performed using the QuantiFast SYBR Green PCR Kit according to the manufacturer's instructions, and each reaction was performed in triplicate using the Roche LightCycler 96 system. The results were analyzed with absolute quantification using the second derivative maximum and the $2^{-\Delta Ct}$ method. Each ChIP experiment was repeated at least three times. Statistical analysis was performed using Student's unpaired $t$-test. The error bars represent s.d.

## Chromatin fractionation

The chromatin-binding assay was performed as described previously[16,55]. Briefly, native yeast protein extract was prepared from 50 ml with an $OD_{600} = 1$ of logarithmically growing culture by treating collected cells with zymolyase to produce spheroplasts and disrupting them with 1% Triton X-100. The resulting whole-cell extract was carefully applied on top of a 30% sucrose cushion of equal volume and centrifuged for 30 min at 20,000$g$ at 4 °C. The supernatant containing the soluble protein fraction was carefully collected from the top of the cushion, sucrose was aspirated, and the pellet containing the chromatin fraction was resuspended in HU sample buffer for subsequent SDS–PAGE and western blot analysis.

## Cell lines and general techniques

Cell lines used in this study are listed in Supplementary Table 2. mRNA isolation, reverse transcription PCR, western blotting and cohesion analysis were performed as previously described[58]. The Nikon NIS-Elements platform was used for the cohesion assay in DT40 cells.

## Cell culturing

TK6 cells were cultured at 37 °C in RPMI-1640 medium supplemented with 10% horse serum, 2 mM L-glutamine, penicillin–streptomycin mix and 1.8 mM sodium pyruvate. DT40 cells were cultured at 39.5 °C in DMEM/F-12 GlutaMAX supplement medium supplemented with 10% FBS, 2% chicken serum, penicillin–streptomycin mix and 10 μM 2-mercaptoethanol in the presence or absence of 500 μM auxin. To plot growth curves, each cell line was cultured in three different wells of 24-well plates and passaged every 24 h. Cell numbers were determined at each time point by flow cytometry. The BD Accuri C6 Plus was used for flow cytometry analysis. For cell cycle synchronization of TK6 cells, cells were treated with 100 ng ml$^{-1}$ nocodazole for 12 h and released into drug-free medium.

## Plasmid construction and transfection

To generate the chicken *NIPBL* expression construct, the sequence for the 3× HA tag was inserted into the EcoRV locus of the pBACT-Puro vector[59]. Subsequently, full-length *NIPBL* cDNA was amplified using primers 5′-GCATGCGGCCGCTAATGGGGATATGCCTCATGTTCC-3′ (NotI) and 5′-GCATGGTACCTTAGCTCGAAGTTCCATCCTTGG-3′ (KpnI) and cloned into the pBACT-3xHA-Puro vector. The construct was linearized with FspI before transfection. For engineering

minichromosome 2, the *GFP* cassette of the telomere-seeding vector[45] was replaced with the *mCherry* cassette. Otherwise, the plasmids used for the minichromosome-loss assay and the transfection method are previously described[45]. To generate the *NIPBLw-EGFP* knockin construct, the homology arm was amplified using primers 5′-AAAGTCGACTGCTGGATAGCGAAGATGGAGAAG-3′ (SalI) and 5′-AAAGCGGCCGCTTCTGCTGGTGCAGATTTCTGTG-3′ (NotI) and ligated into the pLoxP vector[60]. Subsequently, the *Puro-GFP* cassette was cloned into the endogenous BamHI site in the middle of the homology arm. The construct was linearized with NotI before transfection. The *SMAD7-Ecogpt* knockin construct was made by cloning the homology arm amplified by primers 5′-AAAGTCGACcCTTAGGGATGGAGTGGGGCATCCAG-3′ (SalI) and 5′-AAAGCGGCCGCCCATCATGTCATTGGGTGCTTAGG-3′ (NotI) into the pLoxP vector. The *Ecogpt* cassette was then cloned into the endogenous BamHI site. The construct was linearized with NotI before transfection. To truncate the sequence for the last 195 amino acids of NIPBL on chromosome Z and fuse *EGFP*, 2.2 kb of the homology arm downstream of the sequence for NIPBL[E2588] was amplified by using primers 5′-GTACGTCGACCAAACGCAGGAAGAGCCACTG-3′ (SalI) and 5′-GTACGCTAGCTTCAGTGATCTTGGAATTAGCCATATCAC-3′ (NheI) and cloned into the pEGFP-cFLP-Eco vector[59]. The plasmid was linearized with AflII before transfection. For adding EGFP at the C terminus of NIPBL, 2 kb of the homology arm upstream of the stop codon was amplified with the primers 5′-GCATGTCGACCAGGAAGACAGGAGTGCATTTCCATC-3′ (SalI) and 5′-GCATACTAGTGCTCGAAGTTCCATCCTTGGC-3′ (SpeI) and cloned into the pEGFP-cFLP-Eco vector. The plasmid was linearized with NheI before transfection.

## Drug-sensitivity assay

To assess drug sensitivity, $1 \times 10^4$ cells were cultured in 24-well plates containing various concentrations of nocodazole in 1 ml of medium in duplicate. Cell viability was assessed after 48 h with the CellTiter-Glo assay following the manufacturer's protocol. Percentage survival was determined by considering the luminescent intensity of untreated cells as 100%. The Thermo Scientific SkanIt microplate reader was used for the CellTiter-Glo assay.

## Co-immunoprecipitation in TK6 and DT40 cells

For immunoprecipitating NIPBL in TK6 cells, $1 \times 10^7$ cells were lysed in 0.5 ml lysis buffer (20 mM Tris-HCl, pH 7.4, 150 mM NaCl, 5 mM MgCl₂, 0.5% NP-40, 10% glycerol, 20 mM *N*-ethylmaleimide, 1 mM PMSF, 1× cOmplete cocktail, 50 U ml⁻¹ benzonase). Lysates were then rotated at 4 °C for 1 h and at 37 °C for 10 min. After centrifugation (21,000*g*, 4 °C, 20 min), supernatants were incubated with 1 µg anti-NIPBL (A301-779A, Bethyl) and 3 mg Dynabeads Protein A for 2 h. Beads were then washed with 1 ml wash buffer (20 mM Tris-HCl, pH 7.4, 200 mM NaCl, 5 mM MgCl₂, 0.5% NP-40, 10% glycerol, 20 mM *N*-ethylmaleimide, 1 mM PMSF, 1× cOmplete cocktail) four times and incubated with 30 µl of 1× Laemmli buffer at 95 °C for 15 min for elution. To immunoprecipitate 3HA–NIPBL in DT40 cells, $1 \times 10^7$ cells were lysed in 1 ml lysis buffer (20 mM Tris-HCl, pH 7.4, 150 mM NaCl, 5 mM MgCl₂, 0.5% NP-40, 10% glycerol, 20 mM *N*-ethylmaleimide, 1 mM PMSF, 1× cOmplete cocktail). Following 10 min of incubation on ice, lysates were sonicated (10%, 12 s, three cycles) to solubilize the chromatin. After centrifugation (21,000*g*, 4 °C, 5 min) supernatants were incubated with 50 µl Pierce Anti-HA Magnetic Beads overnight. Beads were then washed with 1 ml wash buffer (20 mM Tris-HCl, pH 7.4, 200 mM NaCl, 5 mM MgCl₂, 0.5% NP-40, 10% glycerol, 20 mM *N*-ethylmaleimide, 1 mM PMSF, 1× cOmplete cocktail) four times and incubated with 30 µl of 1× Laemmli buffer at 95 °C for 15 min for elution. Immunoprecipitation samples were analyzed by standard western blotting techniques. Antibodies used were the following: anti-HA (1:1,000, 12158167001, Roche), anti-GAPDH (1:1,000, sc-47724, Santa Cruz Biotechnology), anti-GFP (1:500, sc-9996, Santa Cruz Biotechnology), anti-MAU2 (1:1,000, ab183033, Abcam), anti-MCM7 (1:500, sc-9966, Santa Cruz Biotechnology), anti-miniAID (1:1,000, M214-3, MBL), anti-NIPBL (1:1,000, A301-779A, Bethyl), anti-NIPBL (1:200, sc-374625, Santa Cruz Biotechnology), anti-PCNA (1:2,000, sc-25280, Santa Cruz Biotechnology) and anti-SMC3 (1:1,000, gift from A. Losada).

## Minichromosome-loss assay

Minichromosome 2 was engineered as previously described[45] with small modifications. The *GFP* expression unit of the telomere-seeding vector targeting *TPK1* was replaced with the *mCherry* expression unit. Cells were cultured in medium containing L-histidinol (1 mg ml⁻¹) and puromycin (0.5 µg ml⁻¹) for 48 h before starting the experiment to exclude cells that had already lost minichromosome 2. The percentage of mCherry-negative cells was determined by flow cytometry every 48 h. Ten thousand cells were analyzed for each condition. The flow cytometry gating strategy and representative images for the minichromosome-loss assay are shown in Supplementary Fig. 1.

## BrdU–ChIP–slot–western technique

The BrdU–ChIP–slot–western technique was performed as previously described[44] with minor modifications. A total of $1 \times 10^7$ cells were pulse labeled with 20 µM BrdU for 20 min and cross-linked with 1% paraformaldehyde for 10 min at room temperature. Cross-linking was then quenched with 125 mM glycine for 5 min at room temperature. Following centrifugation (900*g*, 4 °C, 10 min), pellets were washed with 10 ml of cold PBS twice. Subsequently, pellets were resuspended in 0.5 ml of cold FA140 buffer (50 mM HEPES-KOH, pH 7.5, 140 mM NaCl, 1 mM EDTA, pH 8.0, 1% Triton X-100, 0.1% sodium deoxycholate, 1 mM PMSF, 1× cOmplete cocktail) and sonicated to shear chromatin (level 6, 10 s, six cycles). Following sonication, lysates were centrifuged (21,000*g*, 4 °C, 10 min). Supernatants were then incubated with 1 µg anti-SMC3 (a gift from A. Losada) and 1 mg Dynabeads Protein A at 4 °C overnight. Next, 0.1 ml of the supernatant was taken as 'input' and kept on ice. Beads were washed with 0.5 ml of cold FA140 buffer for 5 min and with 0.5 ml of cold FA500 buffer (50 mM HEPES-KOH, pH 7.5, 500 mM NaCl, 1 mM EDTA, pH 8.0, 1% Triton X-100, 0.1% sodium deoxycholate, 1 mM PMSF, 1× cOmplete cocktail) for 5 min. Subsequently, beads were washed with 0.5 ml of cold LiCl buffer (10 mM Tris-HCl, pH 8.0, 250 mM LiCl, 0.5% NP-40, 0.5% sodium deoxycholate, 1 mM PMSF, 1× cOmplete cocktail). Following the wash with the LiCl buffer, beads were incubated with 0.2 ml elution buffer (1% SDS, 100 mM NaHCO₃) at 25 °C for 15 min twice for elution and collected together. Next, 0.3 ml elution buffer was added to the 'input' samples. NaCl (13.2 µl of 5 M) was added to the eluted 'immunoprecipitated' and 'input' samples and incubated at 65 °C for 5 h to reverse the cross-links. Next, 1 ml ethanol was added, and samples were kept at −80 °C overnight and finally centrifuged (21,000*g*, 4 °C, 15 min). Pellets were washed with 0.8 ml of 70% ethanol once and air dried at 37 °C for 10 min. Pellets were then resuspended in 90 µl Milli-Q water, and 2 µl of 10 mg ml⁻¹ RNase A was added and incubated at 37 °C for 30 min. After RNase A treatment, 10 µl of 10× proteinase K buffer (100 mM Tris-HCl, pH 8.0, 50 mM EDTA, pH 8.0, 5% SDS) and 1 µl of 20 mg ml⁻¹ proteinase K were added, and then samples were incubated at 42 °C for 1 h. Subsequently, DNA was purified using the PCR Purification Kit (Qiagen) following the manufacturer's protocol and eluted with 50 µl Milli-Q water. The DNA concentration was measured using the Qubit dsDNA HS Assay Kit (Invitrogen), and the concentration was adjusted to 30 ng µl⁻¹ for 'input' and 0.25 ng µl⁻¹ for immunoprecipitated fractions in a volume of 50 µl. Samples were then denatured by adding 125 µl of 0.4 M NaOH and incubated for 30 min at room temperature. Following denaturation, 175 µl of 1 M Tris-HCl, pH 6.8 was added for neutralization, and samples were kept on ice. Denatured DNA was then blotted onto a nitrocellulose membrane prewet with 20× SSC and cross-linked using a UV chamber (125 mJ, Bio-Rad). The membrane was then subjected to antibody reaction following the

same procedure for western blotting. The Bio-Rad ChemiDoc Touch imaging system was used for western blot and slot blot acquisition. Mouse monoclonal anti-BrdU (1:500, 347580, BD Biosciences) was used for BrdU detection.

## Protein structure prediction using AlphaFold2 and AlphaFold-Multimer software

The structures of individual proteins and protein complexes were predicted using the Google DeepMind AlphaFold2 and AlphaFold-Multimer systems[31,41] available on Google's cloud computing software with the AlphaFold Colab notebook service (https://colab.research.google.com/github/deepmind/alphafold/blob/main/notebooks/AlphaFold.ipynb). Analysis of protein structures was performed using the PyMOL Molecular Graphics System, version 1.7.4.5 (Schrödinger).

## Statistics and reproducibility

No samples were measured repeatedly for statistical analysis. Two-tailed unpaired Student's *t*-test was performed as indicated in the figure legends using GraphPad Prism (version 9). Sample sizes and statistical tests used are specified in the figure legends. All experimental findings were confirmed by independent repetitions. No data were excluded from the analyses. No statistical method was used to predetermine sample size. Data shown in Figs. 2c,d, 4a,b and 5a,d and Extended Data Figs. 1d, 2b, 3a, 5a–c,e, 7e, 8a,b,e,f and 10c,d,g were confirmed in at least two independent experiments; data shown in Figs. 2f, 4c–f, 5e and 6a–d and Extended Data Fig. 10e,f were confirmed in at least three independent experiments.

## Reporting summary

Further information on research design is available in the Nature Portfolio Reporting Summary linked to this article.

## Data availability

The authors declare that the data supporting the findings of this study are available within the article. Source data are archived at the IFOM ETS, the AIRC Institute of Molecular Oncology or the Department of Chemistry at Tokyo Metropolitan University. The following publicly available datasets were used in the study: PDB 6ZZ6, 6WGE and 6YUF. Source data are provided with this paper.

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

## Acknowledgements

We thank S. P. Bell (MIT, Cambridge), S. Hahn (Fred Hutchinson Cancer Center), S. Jentsch (MPIB, Martinsried), A. Losada (CNIO, Madrid), K .A. Nasmyth (University of Oxford), H. Sasanuma (TMiMS, Tokyo), K. Shirahige (University of Tokyo) and L. Warfield (Fred Hutchinson Cancer Center) for sharing reagents; S. Barozzi, F. Casagrande and M. Garre for help with microscopy; and F. Uhlmann for communicating unpublished results. This work was supported by the Italian Association for Cancer Research (IG 18976, IG 23710) and European Research Council (consolidator grant 682190) grants to D.B., an EMBO long-term fellowship (ALTF 561-2014), an AIRC–Marie Curie Actions—COFUND iCARE fellowship and a Fondazione Umberto Veronesi fellowship to I.P. R.K. received support from Japanese Society for the Promotion of Science (JSPS) KAKENHI (22K15040) and was partly supported by an AIRC 3-year fellowship 'Mari e Valeria Rindi' (Rif.22403). T.A. received support from JSPS KAKENHI (20K06760, 22H05072). The laboratory of K.H. was supported by Tokyo Metropolitan Government Advanced Research (grant number R3-2), JSPS KAKENHI (JP21K19235, JP20H04337, 19KK0210), the Yamada Science Foundation, the Novartis Foundation, the Uehara Memorial Foundation and the Takeda Science Foundation.

## Author contributions

D.B., R.K. and I.P. conceived the study. R.K., I.P. and T.A. performed experiments. R.K., I.P., T.A., K.H. and D.B. analyzed data. R.K. and I.P. constructed the figures. D.B., R.K. and I.P. wrote the manuscript.

## Competing interests

The authors declare no competing interests.

## Additional information

**Extended data** is available for this paper at https://doi.org/10.1038/s41594-023-01064-x.

**Correspondence and requests for materials** should be addressed to Ivan Psakhye or Dana Branzei.

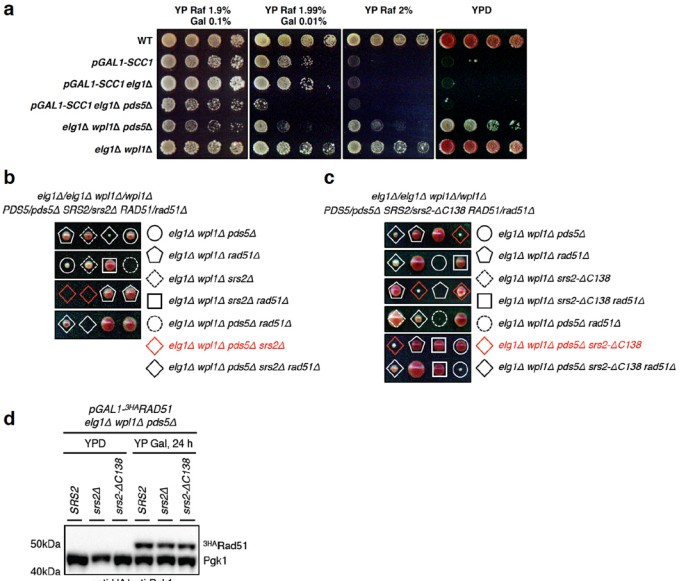

**Extended Data Fig. 1 | Elevated PCNA levels in *elg1Δ* do not promote SCC through Srs2 recruitment, but bring another factor to support viability of *pds5Δ* cells. a**, Overexpression of *SCC1* from galactose-inducible *pGAL1* promoter (Gal 0.1%) supports viability of *elg1Δ pds5Δ* cells, whereas low *SCC1* expression (Gal 0.01%) that supports viability of *elg1Δ* cells does not permit the growth of the *elg1Δ pds5Δ* mutant. Spotting of 1:7 serial dilutions on indicate plates containing glucose (YPD), raffinose (Raf) and galactose (Gal). **b-c**, Tetrad dissection analysis of the indicated diploid strains. Loss of Srs2 in *elg1Δ wpl1Δ pds5Δ* cells results in lethality that can be rescued by Rad51 deletion (**b**), whereas expression of the *srs2-ΔC138* mutant unable to interact with PCNA does not cause lethality in *elg1Δ wpl1Δ pds5Δ* cells (**c**). **d**, Control of the expression of N-terminally 3HA-tagged Rad51 recombinase from the galactose-inducible promoter *pGAL1* in the indicated backgrounds after 24 h shift from glucose-containing YPD to 2% galactose-containing YP Gal media.

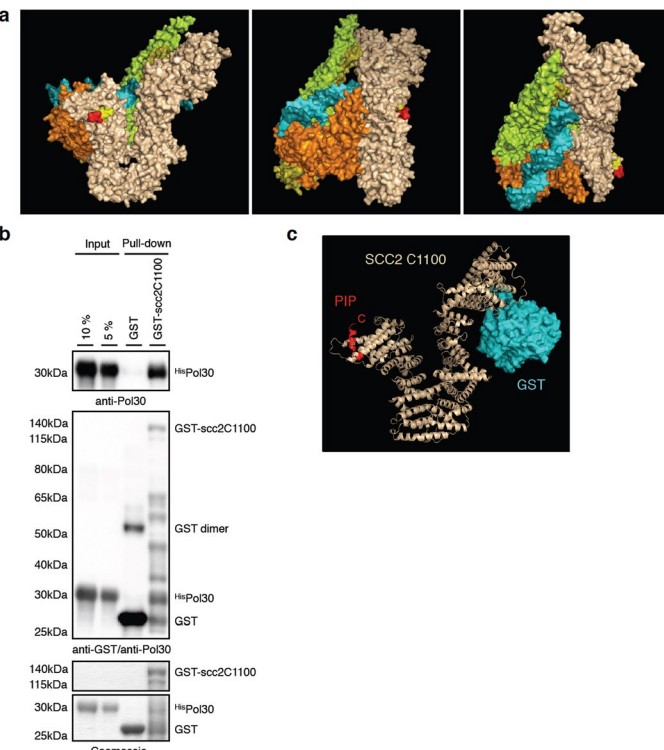

**Extended Data Fig. 2 | Yeast cohesin loader Scc2 interacts with PCNA in vitro.**
**a**, Cryo-EM structure of the budding yeast cohesin-Scc2-DNA complex (PDB
6ZZ6) with C-terminal residues 1466-1470 and 1471-1475 of Scc2 highlighted
yellow and red, respectively. The last 18 residues (1476-1493) of Scc2 containing
consensus PIP motif are not resolved in the structure due to mobility of the C
terminus. The DNA is colored blue, Smc1 – orange, Smc3 – green, Scc1 – olive.

**b**, GST fusion of Scc2 (amino acids 394-1493, GST-scc2C1100) that lacks its
largely unstructured N-terminal 393 residues necessary for binding to Scc4,
interacts with yeast PCNA ($^{His}$Pol30) in vitro. **c**, AlphaFold2 prediction of the
GST-scc2C1100 structure with N-terminal GST highlighted blue and C-terminal
flexible PIP highlighted red.

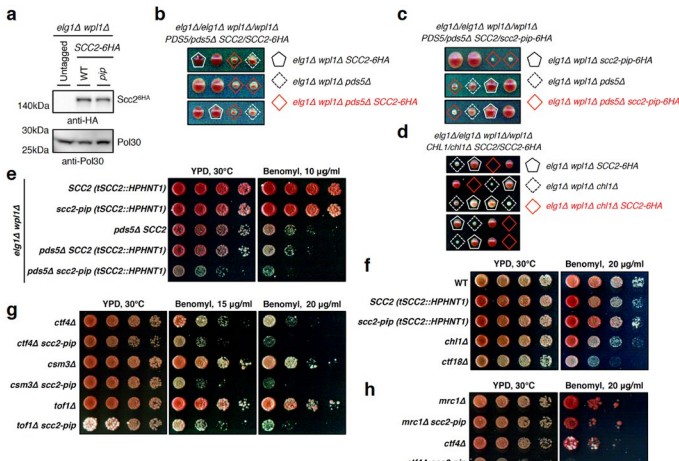

**Extended Data Fig. 3 | The *scc2-pip* mutant shows additive cohesion defects with replisome mutants of the cohesin conversion, but not de novo loading pathway. a**, Protein levels of C-terminally 6HA-tagged Scc2 and its PIP mutant variant in *elg1Δ wpl1Δ* cells. PCNA (yeast Pol30) served as loading control. **b-d**, Tetrad dissection analysis of the indicated diploid strains. C-terminal 6HA-tagging of Scc2 results in the slow-growth phenotype in *elg1Δ wpl1Δ pds5Δ* (**b**), and lethality in *elg1Δ wpl1Δ chl1Δ* (**d**) background. **e**, *scc2-pip* mutant

combined with *elg1Δ wpl1Δ pds5Δ* results in slow-growth phenotype and strong sensitivity to benomyl. Strains with disruption of the *SCC2* terminator by selection marker (*tSCC2::HPHNT1*) without mutating the *SCC2* PIP served as controls. **f-h**, The *scc2-pip* mutant, while on its own not sensitive to microtubule poison (**f**), shows strong sensitivity to benomyl when combined with replisome mutants of the cohesin conversion pathway *ctf4Δ*, *csm3Δ*, and *tof1Δ* (**g**), but not with the de novo cohesin loading pathway mutant *mrc1Δ* (**h**).

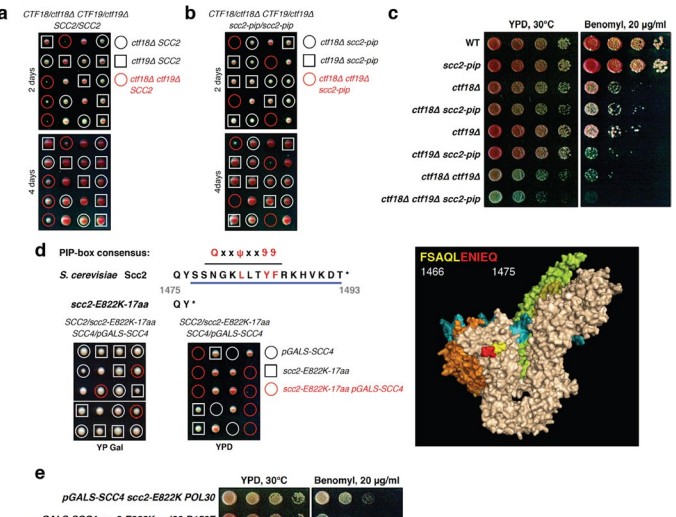

**Extended Data Fig. 4 | The *scc2-pip* mutant shows additive cohesion defects with *ctf19Δ*, the mutant of kinetochore receptor for cohesin loader, and is synthetic lethal with *scc4Δ*. a-b,** Tetrad dissection analysis of the indicated diploid strains. The *scc2-pip* mutant shows synthetic growth defect when combined with *ctf18Δ ctf19Δ* (**b**) in comparison with *ctf18Δ ctf19Δ* double mutant (**a**). **c,** Similar to *ctf18Δ*, the *scc2-pip* mutant shows additive sensitivity to benomyl when combined with *ctf19Δ*. **d,** Tetrad dissection analysis of the

*SCC2/scc2-E822K-17aa SCC4/pGALS-SCC4* diploid strain on galactose- (YP Gal) and glucose-containing (YPD) plates. *scc2-E822K-17aa* mutant lacking the last 17 residues that harbor consensus PIP motif cannot rescue the lethality of cells expressing *SCC4* from the galactose-inducible *pGALS* promoter when tetrads are dissected on YPD plates. **e,** The disassembly-prone PCNA mutant *pol30-D150E* shows additive sensitivity to benomyl when combined with *scc4 scc2-E822K* double mutant.

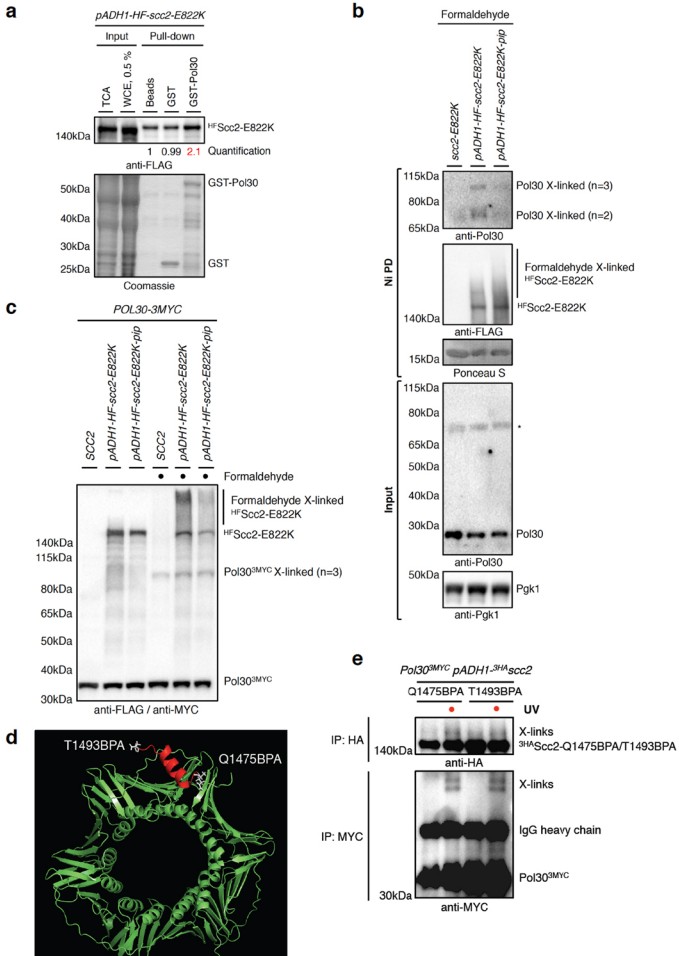

**Extended Data Fig. 5 | Yeast cohesin loader Scc2 interacts with PCNA in vivo.**
**a**, GST pull-down using recombinant GST-Pol30 (yeast PCNA) in lysates from yeast cells expressing N-terminally 7His8FLAG (HF)-tagged Scc2-E822K from a strong constitutive promoter *pADH1*. **b**, Ni-NTA pull-down (Ni PD) of HF-tagged Scc2-E822K or its PIP mutant under denaturing conditions from yeast cells after formaldehyde crosslinking. HFScc2-E822K but not its PIP mutant variant binds to cross-linked PCNA species (Pol30 X-linked). PCNA was detected using polyclonal anti-Pol30 antibody (the asterisk indicates a cross-reactive protein), untagged *scc2-E822K* strain was used as control. Ni PD efficiency was assayed using Ponceau S staining (nonspecifically pulled-down protein of ≈15 kDa visualized), Pgk1

served as loading control. **c**, Cross-linked PCNA species (Pol30³ᴹYC X-linked) detected using anti-MYC antibody after formaldehyde crosslinking of yeast cells expressing C-terminally 3MYC-tagged Pol30. **d**, Residues Q1475 and T1493 in the vicinity of Scc2 PIP selected to be replaced with BPA for in vivo cross-linking experiments. **e**, In vivo BPA cross-linking of Scc2-Q1475BPA and Scc2-T1493BPA to PCNA. Cross-links are detected in cells expressing C-terminally 3MYC-tagged Pol30 (yeast PCNA) and N-terminally 3HA-tagged Scc2-Q1475BPA or Scc2-T1493BPA specifically following UV irradiation. Proteins were isolated by immunoprecipitation.

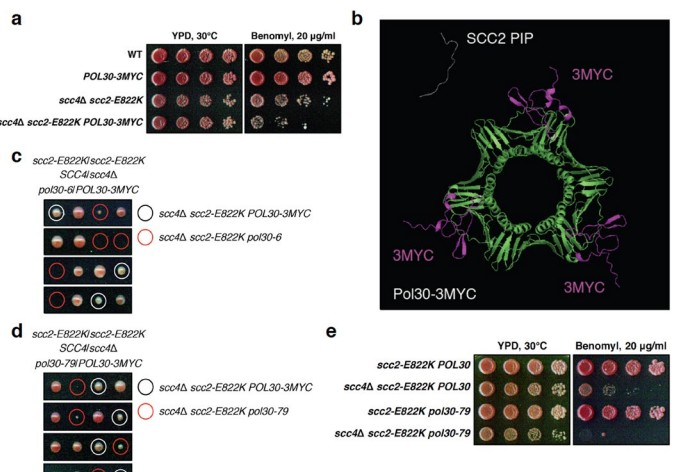

**Extended Data Fig. 6 | The *scc4Δ scc2-E822K* double mutant shows additive defects with PCNA mutants affecting its front face. a**, C-terminally 3MYC-tagged PCNA (*POL30-3MYC*) variant shows additive sensitivity to benomyl when combined with *scc4Δ scc2-E822K* double mutant. **b**, The alphaFold-Multimer prediction of the interaction between Scc2 PIP and C-terminally 3MYC-tagged yeast PCNA (Pol30-3MYC). 3MYC interferes with the interaction by affecting the front face of the PCNA ring. **c-d**, Tetrad dissection analysis of the indicated diploid strains. Mutants causing disruptions of a surface cavity on the front face of the PCNA ring, *pol30-6* (**c**) and *pol30-79* (**d**), show synthetic sickness when combined with *scc4Δ scc2-E822K* double mutant. **e**, *pol30-79* mutant shows additive sensitivity to benomyl when combined with *scc4Δ scc2-E822K* double mutant.

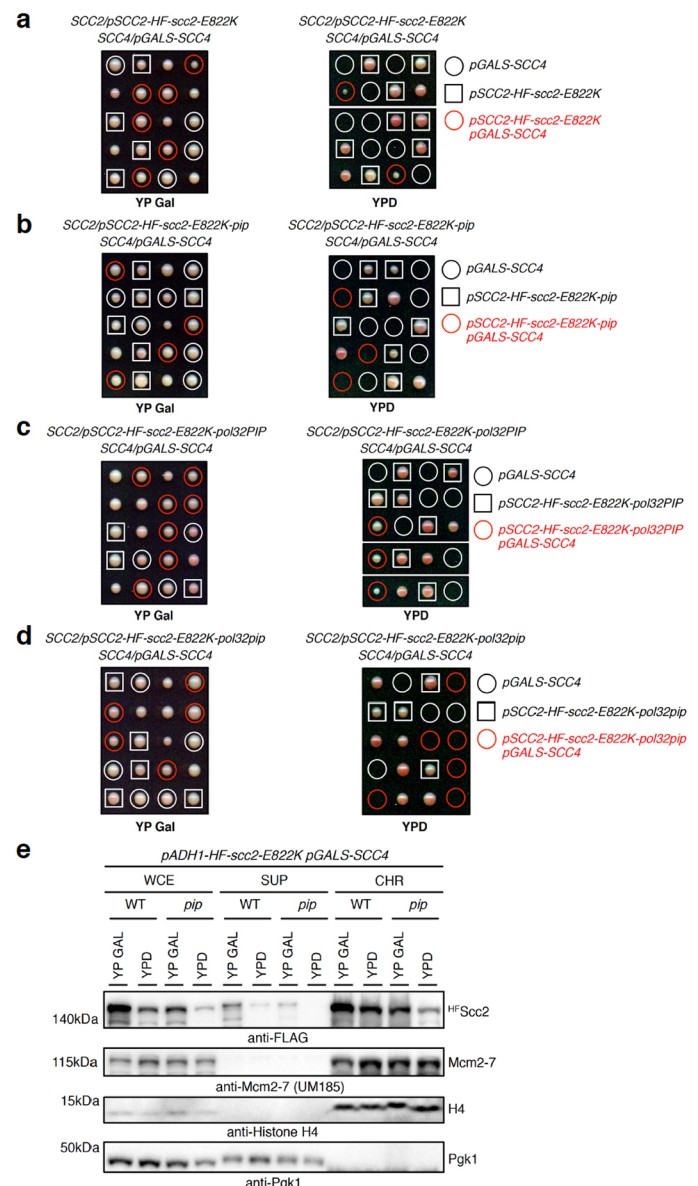

**Extended Data Fig. 7 | Scc4 and PIP of Scc2 are both contributing to chromatin binding of the cohesin loader. a-d**, Tetrad dissection analysis on the indicated diploid strains on galactose- (YP Gal) and glucose-containing (YPD) plates. N-terminally 7His8FLAG (HF)-tagged Scc2-E822K and its various PIP mutants are expressed from endogenous promoter *pSCC2*. Cohesin loaders ^HFScc2-E822K with intact PIP motifs, either endogenous (**a**) or replaced by the PIP of the DNA polymerase δ nonessential subunit Pol32 (**c**), support viability upon *SCC4* shut-off on YPD plates. On the contrary, PIP mutants with conserved residues replaced by alanines (**b**, **d**) fail in providing viability. **e**, Subcellular fractionation of cycling cells expressing HF-tagged Scc2-E822K or its PIP mutant from strong constitutive promoter *pADH1* into soluble supernatant (SUP) and chromatin-enriched (CHR) fractions by centrifugation of the whole cell extract (WCE). Chromatin binding of ^HFScc2-E822K-pip is decreased compared to ^HFScc2-E822K. Used cells had in addition *SCC4* expressed from galactose-inducible *pGALS* promoter allowing to study ^HFScc2-E822K chromatin binding upon *SCC4* shut-off after shift from galactose-containing media (YP GAL) to glucose (YPD). Loss of Scc4 further decreases cohesin loader binding to chromatin. To control chromatin fractionation efficiency, the levels of histone H4, replicative helicase Mcm2-7 and the cytoplasmic/plasma membrane protein Pgk1 were detected in fractions.

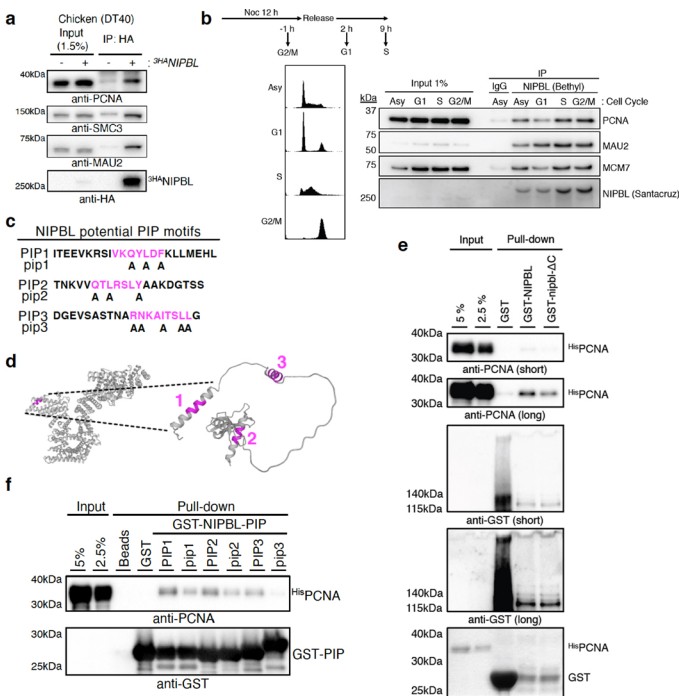

**Extended Data Fig. 8 | Interaction between PCNA and NIPBL through its C-terminal PIP-like motifs in vertebrates. a**, Immunoprecipitation of overexpressed 3HA-NIPBL in DT40 cells. PCNA was co-immunoprecipitated with 3HA-NIPBL. SMC3 and MAU2 are shown as positive control. **b**, Cell cycle synchronization of human TK6 cells followed by immunoprecipitation of NIPBL. More PCNA is co-immunoprecipitated with NIPBL in S and G2/M compared to G1 phase and asynchronous cell population. *TP53* knockout cells were used to avoid cell cycle arrest in G1 following release from nocodazole treatment. **c**, Peptides used for GST in vitro pull-down assay are shown. PIP-like motifs (PIP1-3) at NIPBL C-terminus are colored pink. Amino acids substituted to alanine in the mutant versions of PIPs (pip1-3) are indicated. **d**, Structures of human NIPBL solved by cryo-EM (PDB 6WGE; left) and the last 203 residues of chicken NIPBL (2581-2783; right) predicted by AlphaFold2. PIP-like motifs are colored pink. **e**, Fusion of the full-length NIPBL to GST (residues 1-2783; GST-NIPBL) interacts with chicken HisPCNA. Truncation of the last 195 amino acids of NIPBL containing PIP-like motifs (residues 1-2588; GST-nipbl-ΔC) reduces the interaction. **f**, All three PIP-like motifs fused to GST interact with chicken PCNA. Mutating consensus amino acids of PIP motifs weakens the interaction in vitro.

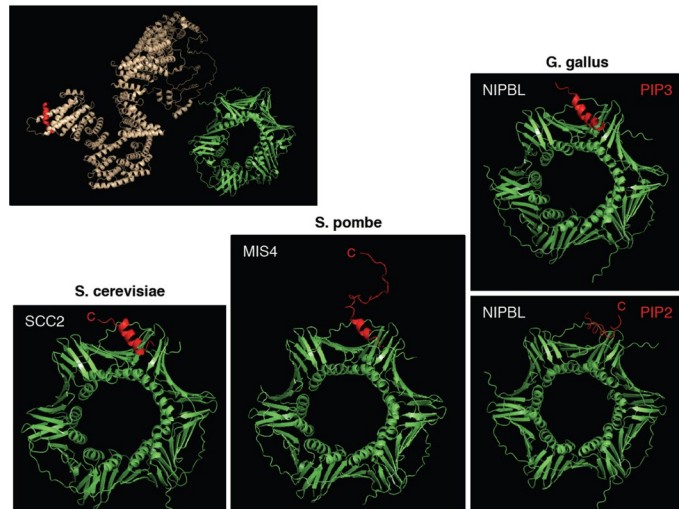

**Extended Data Fig. 9 | The alphaFold-Multimer predictions of the interaction between PCNA and the C-terminal PIPs of cohesin loaders conserved from yeast to vertebrates.** The alphaFold2 prediction of the budding yeast cohesin loader Scc2 next to PCNA ring with its front face positioned towards the flexible C-terminal PIP motif of Scc2 colored red (upper left panel). The interactions of the C-terminal PIPs of the budding yeast S. cerevisiae cohesin loader Scc2 (residues 1476-1493), the fission yeast S. pombe cohesin loader Mis4 (residues 1553-1587), and the chicken G. gallus cohesin loader NIPBL (PIP3 residues 2621-2640; PIP2 residues 2764-2783) with PCNA predicted using alphaFold-Multimer (bottom panels from left to right; C-terminal protein ends are labeled with C). The PIPs are located within flexible C-terminal tails of cohesin loaders, the cryo-EM structures of which were not resolved previously (PDB 6ZZ6 – S. cerevisiae; PDB 6YUF – S. pombe; PDB 6WGE – H. sapiens) due to their mobility.

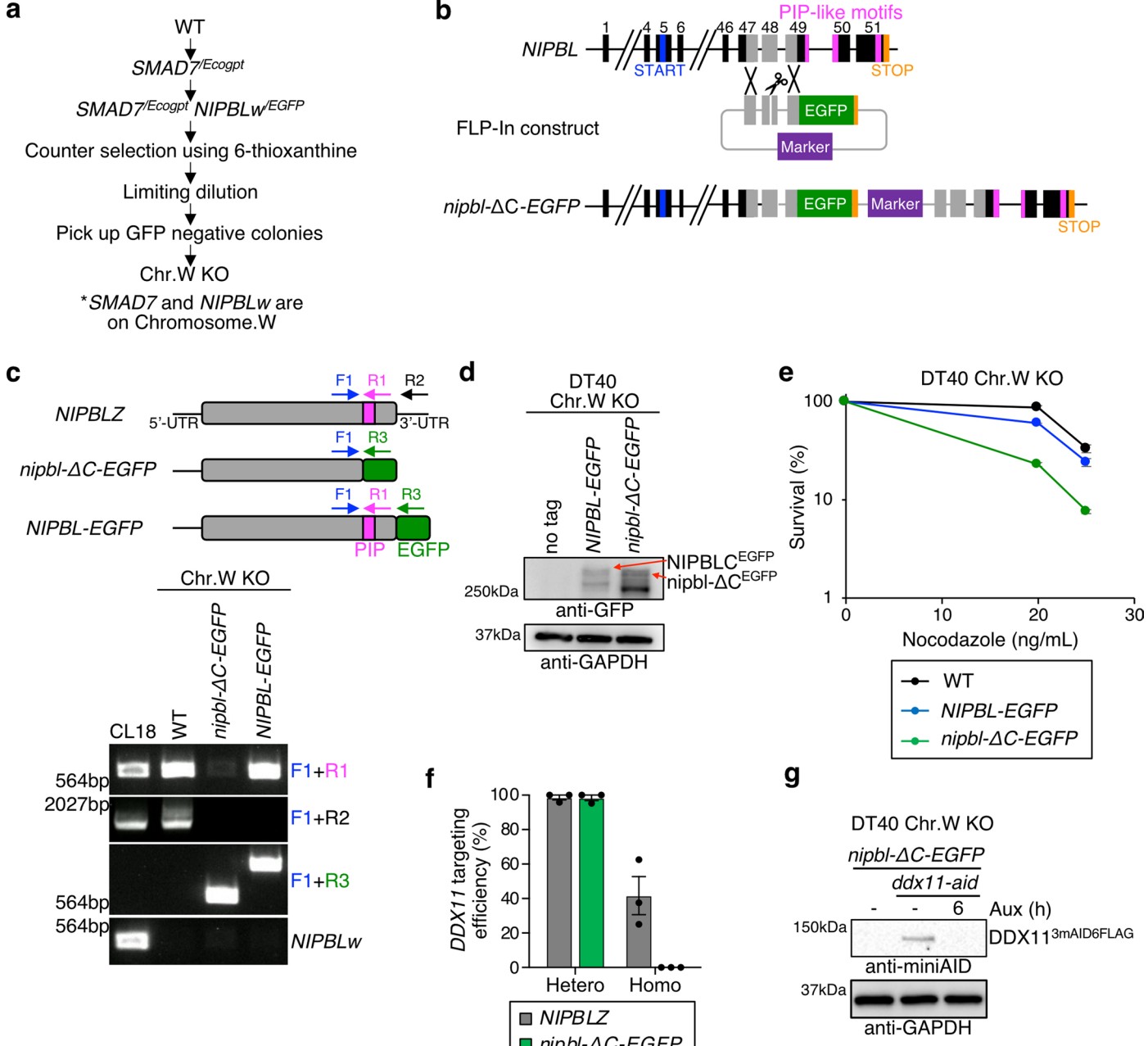

**Extended Data Fig. 10 | Establishment of *nipbl-ΔC-EGFP ddx11-aid* conditional mutant. a**, Scheme of chromosome W knockout establishment in DT40 cells. **b**, Schematic representation of *NIPBL* gene locus on chromosome Z, and FLP-In construct replacing the last 195 residues of NIPBL with EGFP. (Closed boxes) exons; (gray boxes) homology arm; (blue box) start codon; (pink boxes) PIP-like motifs; (yellow boxes) stop codon; (green boxes) EGFP; (purple boxes) selection marker gene. **c**, Reverse-transcription PCR analysis verifying *nipbl-ΔC-EGFP* and *NIPBL-EGFP* cells. Primers designed on *NIPBL* cDNA are shown in the upper panel. **d**, Western blotting analysis validating *nipbl-ΔC-EGFP* and *NIPBL-EGFP* cells. **e**, Sensitivity assay using CellTiter-Glo. *NIPBL-EGFP* cells do not exhibit hypersensitivity to nocodazole treatment unlike *nipbl-ΔC-EGFP* cells. The mean values of three independent experiments ± SD are plotted. **f**, Targeting efficiency of *DDX11* knockout was determined by genomic PCR. At least 18 clones were screened in each experiment. The mean values three, n=3, independent experiments ± SEM are plotted. **g**, Western blotting analysis confirming the depletion of DDX11 tagged with AID, 6 h after auxin addition.

# Reporting Summary

## Statistics

For all statistical analyses, confirm that the following items are present in the figure legend, table legend, main text, or Methods section.

| n/a | Confirmed | |
|---|---|---|
| ☐ | ☒ | The exact sample size (*n*) for each experimental group/condition, given as a discrete number and unit of measurement |
| ☐ | ☒ | A statement on whether measurements were taken from distinct samples or whether the same sample was measured repeatedly |
| ☐ | ☒ | The statistical test(s) used AND whether they are one- or two-sided<br>*Only common tests should be described solely by name; describe more complex techniques in the Methods section.* |
| ☒ | ☐ | A description of all covariates tested |
| ☒ | ☐ | A description of any assumptions or corrections, such as tests of normality and adjustment for multiple comparisons |
| ☐ | ☒ | A full description of the statistical parameters including central tendency (e.g. means) or other basic estimates (e.g. regression coefficient) AND variation (e.g. standard deviation) or associated estimates of uncertainty (e.g. confidence intervals) |
| ☐ | ☒ | For null hypothesis testing, the test statistic (e.g. *F*, *t*, *r*) with confidence intervals, effect sizes, degrees of freedom and *P* value noted<br>*Give P values as exact values whenever suitable.* |
| ☒ | ☐ | For Bayesian analysis, information on the choice of priors and Markov chain Monte Carlo settings |
| ☒ | ☐ | For hierarchical and complex designs, identification of the appropriate level for tests and full reporting of outcomes |
| ☒ | ☐ | Estimates of effect sizes (e.g. Cohen's *d*, Pearson's *r*), indicating how they were calculated |

*Our web collection on statistics for biologists contains articles on many of the points above.*

## Software and code

Policy information about availability of computer code

| Data collection | BIORAD Image Lab Version 5.2.1 for Western Blot acquisition<br>BIORAD ChemiDoc Touch Imaging System for Western Blot and Slot Blot<br>BD Accuri C6 Plus for FACS sample collection<br>Roche LightCycler 96 version 1.1.0.1320 for qPCR<br>DeltaVision microscope (Applied Precision) for premature sister chromatid separation assay in budding yeast<br>Nikon NIS-Elements for cohesion assay in DT40 cells<br>SkanIt 6.1 for CellTiter-Glo assay |
|---|---|
| Data analysis | BIORAD Image Lab Version 5.2.1 for quantification of Western Blots<br>BD Accuri C6 Plus for FACS analysis<br>GraphPad Prism Version 9.0.2 (161) for ChIP-qPCR analysis<br>ImageJ 1.49v for image analysis of premature sister chromatid separation assay in budding yeast<br>Google DeepMind alphaFold2 and alphaFold-Multimer, v2.3.2<br>PyMOL molecular graphics system, v1.7.4.5 Schrödinger, LLC |

For manuscripts utilizing custom algorithms or software that are central to the research but not yet described in published literature, software must be made available to editors and reviewers. We strongly encourage code deposition in a community repository (e.g. GitHub). See the Nature Portfolio guidelines for submitting code & software for further information.

## Data

Policy information about availability of data

All manuscripts must include a data availability statement. This statement should provide the following information, where applicable:

- Accession codes, unique identifiers, or web links for publicly available datasets
- A description of any restrictions on data availability
- For clinical datasets or third party data, please ensure that the statement adheres to our policy

The authors declare that the data supporting the findings of this study are available within the article. Source data are provided with this paper, and archived at the IFOM ETS, the AIRC Institute of Molecular Oncology, or the Department of Chemistry at Tokyo Metropolitan University. The following publicly available datasets were used in the study: PDB identifiers 6ZZ6, 6WGE, and 6YUF.

## Human research participants

Policy information about studies involving human research participants and Sex and Gender in Research.

| Reporting on sex and gender | NA |
| --- | --- |
| Population characteristics | NA |
| Recruitment | NA |
| Ethics oversight | NA |

Note that full information on the approval of the study protocol must also be provided in the manuscript.

# Field-specific reporting

Please select the one below that is the best fit for your research. If you are not sure, read the appropriate sections before making your selection.

☒ Life sciences ☐ Behavioural & social sciences ☐ Ecological, evolutionary & environmental sciences

For a reference copy of the document with all sections, see nature.com/documents/nr-reporting-summary-flat.pdf

# Life sciences study design

All studies must disclose on these points even when the disclosure is negative.

| Sample size | No statistical methods were used to predetermine sample size, as this study did not include animal models or human participants. Sample size was determined based on the established standards in the field and experiments to obtain statistical significance and reproducibility. We used 10 000 cells for each flow cytometry analysis point (Abe et al., Sci Rep, 2021), 10 million cells for each ChIP sample (Psakhye et al., Mol Cell, 2019), over 100 nuclei analysis for cohesion assays in DT40 cells (Kawasumi et al, Genes Dev, 2021), over 200 cells per each yeast strain for premature sister chromatid separation assay in budding yeast (Michaelis et al., Cell, 1997). |
| --- | --- |
| Data exclusions | No data were excluded. |
| Replication | All experimental findings were reliably reproduced as indicated in the figure legends. All experiments were repeated at least 2 times to ensure reproducibility. We have not experienced cases of non-reproducible data in this study. At least three biological replicates were included for ChIP-qPCR and cohesion assays. The mean values of at least 3 experiments were calculated and shown with corresponding SD or SEM. Statistical analyses were performed using unpaired two-tailed Student's t-test. |
| Randomization | No randomization was done because this study did not involve animals or human participants. Samples were organized into groups based on treatment and genotype. Appropriate controls were included in all experiments. |
| Blinding | Before each experiment, the strains that were used were given numbers instead of the genotype or condition and the numbers were then connected to the yeast strains or cell line genotype only after analysis. |

# Reporting for specific materials, systems and methods

We require information from authors about some types of materials, experimental systems and methods used in many studies. Here, indicate whether each material, system or method listed is relevant to your study. If you are not sure if a list item applies to your research, read the appropriate section before selecting a response.

## Materials & experimental systems

| n/a | Involved in the study |
|---|---|
| ☐ | ☒ Antibodies |
| ☐ | ☒ Eukaryotic cell lines |
| ☒ | ☐ Palaeontology and archaeology |
| ☒ | ☐ Animals and other organisms |
| ☒ | ☐ Clinical data |
| ☒ | ☐ Dual use research of concern |

## Methods

| n/a | Involved in the study |
|---|---|
| ☒ | ☐ ChIP-seq |
| ☐ | ☒ Flow cytometry |
| ☒ | ☐ MRI-based neuroimaging |

## Antibodies

| Antibodies used | Mouse monoclonal anti-FLAG antibody (1:2000, clone M2; F3165) was purchased from Sigma-Aldrich. Mouse monoclonal anti-Pgk1 antibody (1:2000, clone 22C5D8; cat # 459250) was obtained from Thermo Fisher Scientific. Mouse monoclonal anti-HA (1:2000, clone F-7; sc-7392) and anti-PCNA (1:2000, clone F-2; sc-25280) antibodies were from Santa Cruz Biotechnology, as well as normal mouse IgG. Rabbit polyclonal anti-Pol30 antibody (1:2000; GTX64144) was purchased from Gene Tex. Mouse monoclonal anti-c-MYC antibody (1:2000, clone 9E10) and rabbit polyclonal anti-GST antibody (1:2000) were produced in house. Rabbit polyclonal anti-Histone H4 antibody (1:2000; ab7311) was obtained from Abcam. Mouse monoclonal anti-acetyl-Smc3 antibody (1:2000) was a gift from Katsuhiko Shirahige (Borges et al., 2010). Rabbit polyclonal anti-Mcm2-7 antibody (1:5000; UM185) was a gift from Stephen P. Bell (Bowers et al., 2004). anti-NIPBL antibody (1:1000; A301-779A) was from Bethyl Laboratories. anti-SMC3 antibody (1:1000) was a gift from Ana Losada. anti-BrdU antibody (1:500; 347580) was from BD Biosciences. anti-GAPDH antibody (1:10000; sc-47724), anti-GFP antibody (1:500; sc-9996), anti-MCM7 antibody (1:500; sc-9966), anti-NIPBL (1:200; sc-374625) were purchased from from Santa Cruz Biotechnology. anti-HA antibody (1:1000; 11867423001) was from Roche. anti-MAU2 antibody (1:1000; ab183033) was purchased from Abcam. anti-miniAID antibody (1:1000; M214-3) was from MBL. Anti-rabbit IgG (7074S) and anti-mouse IgG (7076S), HRP-linked antibodies (1:5000) were purchased from Cell Signaling Technology. |
|---|---|
| Validation | All antibodies in this study were used for Western Blot analysis in S. cerevisiae yeast, human TK6 and chicken DT40 samples and the bands for the respective proteins corresponded with the expected size. The application (Western Blot) and species were indicated on the manufacturers websites.<br>We have validated anti-HA antibody in Western Blot by using an S. cerevisiae yeast strain without any tag to confirm specificity. The Pgk1 antibody is commonly used as a loading control in many publications (Gay et al., 2018). |

## Eukaryotic cell lines

Policy information about cell lines and Sex and Gender in Research

| Cell line source(s) | Budding yeast Saccharomyces cerevisiae strains (W303 background), chicken DT40, and human TK6 (Japanese Collection of Research Bioresources Cell Bank; https://cellbank.nibiohn.go.jp) cell lines. |
|---|---|
| Authentication | The yeast strains and chicken DT40 cell lines were confirmed with resistance markers, PCR, sequencing, western blotting wherever relevant. The human TK6 cell lines used were not authenticated. |
| Mycoplasma contamination | All cell lines used were tested negative for mycoplasma contamination. |
| Commonly misidentified lines<br>(See ICLAC register) | No commonly misidentified cell lines were used in the study. |

## Flow Cytometry

### Plots

Confirm that:

☒ The axis labels state the marker and fluorochrome used (e.g. CD4-FITC).

☒ The axis scales are clearly visible. Include numbers along axes only for bottom left plot of group (a 'group' is an analysis of identical markers).

☒ All plots are contour plots with outliers or pseudocolor plots.

☒ A numerical value for number of cells or percentage (with statistics) is provided.

### Methodology

| Sample preparation | Exponentially growing DT40 cells were analyzed by flow cytometry without any preparation procedures. |
|---|---|
| Instrument | BD Accuri C6 Plus Flow Cytometry |
| Software | BD Accuri C6 Plus |

| Cell population abundance | About 70% were living cells, which were tested for mCherry (FL3) signal. |
|---|---|
| Gating strategy | Samples were gated on SSC-A and FSC-A to exclude dead cells and debris. Then living cells (P1) were gated on SSC-A and FL3 to determine the percentage of mCherry negative cells (P2). 10000 of living cells per sample were analysed. |

☒ Tick this box to confirm that a figure exemplifying the gating strategy is provided in the Supplementary Information.

