## [Peer Review File · Nature Structural & Molecular Biology]

Peer Review Information

Manuscript Title: PCNA recruits cohesin loader Scc2/NIPBL to ensure sister chromatid cohesion

Corresponding author name(s): Dana Branzei, Ivan Psakhye

Reviewer Comments & Decisions:

Decision Letter, initial version:

Message: Dear Dr. Branzei,

Thank you again for submitting your manuscript "PCNA recruits cohesin loader Scc2/NIPBL to ensure sister chromatid cohesion". I apologize for the delay in responding, which resulted from the difficulty in obtaining suitable referee reports. Nevertheless, we now have comments (below) from the 2 reviewers who evaluated your paper. In light of those reports, we remain interested in your study and would like to see your response to the comments of the referees, in the form of a revised manuscript.

You will see that all reviewers appreciate the results and find the conclusions timely and of wide interest. There are, however, several comments and suggestions that should be addressed in a revision. Please be sure to address/respond to all concerns of the referees in full in a point-by-point response and highlight all changes in the revised manuscript text file. If you have comments that are intended for editors only, please include those in a separate cover letter.

We expect to see your revised manuscript within 6 weeks. If you cannot send it within this time, please contact us to discuss an extension; we would still consider your revision, provided that no similar work has been accepted for publication at NSMB or published elsewhere.

As you already know, we put great emphasis on ensuring that the methods and statistics reported in our papers are correct and accurate. As such, if there are any changes that should be reported, please submit an updated version of the Reporting Summary along

with your revision.

Reporting Summary:

Please note that all key data shown in the main figures as cropped gels or blots should be presented in uncropped form, with molecular weight markers. These data can be aggregated into a single supplementary figure item. While these data can be displayed in a relatively informal style, they must refer back to the relevant figures. These data should be submitted with the final revision, as source data, prior to acceptance, but you may want to start putting it together at this point.

Data availability: this journal strongly supports public availability of data. All data used in accepted papers should be available via a public data repository, or alternatively, as Supplementary Information. If data can only be shared on request, please explain why in your Data Availability Statement, and also in the correspondence with your editor. Please note that for some data types, deposition in a public repository is mandatory - more information on our data deposition policies and available repositories can be found below:

<https://www.nature.com/nature-research/editorial-policies/reporting-standards#availability-of-data>

[Redacted]

Sincerely,

Carolina Perdigoto, PhD
Chief Editor
Nature Structural & Molecular Biology
orcid.org/0000-0002-5783-7106

Referee expertise:

Referee #1: cohesin and genome org

Referee #2: genome stability, replication

Reviewers' Comments:

Reviewer #1:

Remarks to the Author:

Sister chromatid cohesion generated by the cohesin ring during S-phase is essential to ensure faithful segregation of chromosomes in daughter cells during mitosis. The mechanisms by which cohesin establishes sister chromatid cohesion during S-phase are not fully understood. Pioneering studies suggest that interplay between cohesin and replisome-associated proteins is necessary to ensure tethering of sister DNA in the cohesin ring. However, the precise nature of these interactions remains unknown.

Psakhye et al. present an interesting study describing a conserved role of PCNA in the establishment of sister chromatid cohesion. The overall experiment suggests that PCNA ensures the recruitment of recruits Scc2/NIPBL via its C-terminal. This study addresses an important and interesting problem and may shed light on the establishment of sister chromatid cohesion, a poorly understood process. However, I would recommend acceptance only after addressing the following points:

Major points:

Figure 1.

According to the authors Scc2 (Figure 1a) is recruited by PCNA via a domain located in its C-terminal part. In vitro data showing that PCNA interacts in vitro with the PIP domain of Scc2 are in agreement with this model. Nevertheless, in order to strengthen this hypothesis, I suggest to the authors to test this interaction in vivo by performing for example Co-IP experiments between endogenous proteins (without addition of formaldehyde which could induce artifacts). If the authors do not succeed in revealing this interaction using classical co-IP techniques, they could introduce the photoreactive amino acid Bpa in the pip domain of Scc2 in order to perform site-specific cross-linking assays (PMCID: PMC2483787). It would also be interesting to check Chip-Seq if during the S phase PCNA colocalizes on the DNA with cohesin (and Scc2).

Figure 3

For clarity, I advise the authors to explain why it is relevant to use the scc2-E822K mutant to address the importance of the PIP domain of Scc2 in the recruitment of the loader to chromatin. The same is true for the inactivation of Scc4. In addition, I think it would be interesting and important to analyze the localization of the scc2-pip mutant in a genetic context lacking scc2-E822K. To address this, it will be preferable to use calibrated Chip-Seq technology rather than Chip q-PCR or chromatin fractionation assays that do not allow genome-wide assessment of scc2 recruitment. It will also be interesting to analyze using Chip-Seq effect of the scc2-pip mutant on cohesin localization on chromatin. Since the pip domain is important for Scc2 recruitment, it is expected that the scc2-pip mutant will reduce the amount of cohesin on DNA throughout the genome.

Minor points

I advise authors to include in the introduction key references that have previously suggested an interaction between DNA replication machinery and cohesin:

-Skibbens RV, Corson LB, Koshland D, Hieter P (1999) Ctf7p is essential for sister chromatid cohesion and links mitotic chromosome structure to the DNA replication machinery. *Genes Dev* 13: 307–319.

-Toth A, Ciosk R, Uhlmann F, Galova M, Schleiffer A, et al. (1999) Yeast Cohesin complex requires a conserved protein, Eco1p(Ctf7), to establish cohesion between sister chromatids during DNA replication. *Genes Dev* 13: 320–333.

-Mayer, M.L., Gygi, S.P., Aebersold, R., and Hieter, P. (2001). Identification of RFC(Ctf18p, Ctf8p, Dcc1p): an alternative RFC complex required for sister chromatid cohesion in *S. cerevisiae*. *Mol. Cell* 7, 959–970.

Reviewer #2:

Remarks to the Author:

This manuscript by Psakhye et al. describes a unique role of PCNA in recruiting the cohesin loader, Scc2, during de novo cohesin loading. Cohesion occurs via two mechanisms. The first entails establishment of cohesion during S-phase from cohesins that are already loaded prior to DNA replication, whereas the second requires cohesin loading during S-phase in a manner that depends on PCNA and specific clamp loading complexes. While genetic determinants of this second phase were known, mechanisms were not. The authors identified a PCNA interacting motif, PIP box, in Scc2 to be important for binding PCNA and sister chromatid cohesion. They characterized the functions of the Scc2 PIP box using elegant genetic experiments that reveal the essentiality of the PIP box in the absence of Scc4 and its role in chromatin recruitment. The authors also show the role of PCNA in recruiting Scc2/NIPBL is conserved in higher organisms.

This discovery provides important information towards understanding mechanisms of cohesion. The experiments are designed properly and are well executed. There are, however, a few comments that should be addressed prior to publication.

Major comments:

1) The authors have shown the interaction between Scc2-E822K and PCNA by IP (Extended data 3). But it is unclear when this interaction is observed. For example, is this interaction present only when scc4 is absent. Does scc2 WT also interact with PCNA? Since the de novo pathway is concurrent to DNA replication, a prediction is this association occurs in S phase. The authors should thoroughly investigate the cell cycle dependency of this interaction.

2) PCNA mutants Pol30-6 and pol30-79 could affect several other interactions (such as EcoI) contributing to benomyl sensitivity observed (Extended data 3). IPs detecting interaction differences between pol30-6 and pol30-79 and scc2 would strengthen this data.

3) In Fig. 3a the authors have shown chromatin association of the different Scc2 mutants

and suggested reduced chromatin association of the pip box mutants. But in Fig. 3a total levels (WCE) of the pip mutants appears to be reduced. This suggests overall reduced stability/expression of these mutants rather than a defective chromatin recruitment. Does this affect interpretation of results?

4) Is the contribution of Scc2 PIP and Scc4 in recruiting Scc2 cell cycle dependent?

5) Mutations of single pip boxes in NIPBL seem to affect the interactions with PCNA only partially. A triple mutant of all 3 pip boxes or IPs with NIPBL Δ C would address this issue more clearly.

Minor points

1) What is degrading NIPBL in Fig.4a?

Author Rebuttal to Initial comments

Reviewers' Comments:

Reviewer #1:

Remarks to the Author:

Sister chromatid cohesion generated by the cohesin ring during S-phase is essential to ensure faithful segregation of chromosomes in daughter cells during mitosis. The mechanisms by which cohesin establishes sister chromatid cohesion during S-phase are not fully understood. Pioneering studies suggest that interplay between cohesin and replisome-associated proteins is necessary to ensure tethering of sister DNA in the cohesin ring. However, the precise nature of these interactions remains unknown.

Psakhye et al. present an interesting study describing a conserved role of PCNA in the establishment of sister chromatid cohesion. The overall experiment suggests that PCNA ensures the recruitment of recruits Scc2/NIPBL via its C-terminal. This study addresses an important and interesting problem and may shed light on the establishment of sister chromatid cohesion, a poorly understood process. However, I would recommend acceptance only after addressing the following points:

We are happy that the reviewer acknowledges that the mechanism by which replisome components engender cohesion is not known. Our study uncovers direct recruitment of the cohesin loader to PCNA via a PCNA-interacting protein (PIP) motif present in the C-terminus of Scc2/NIPBL. We are showing that this mechanism is evolutionarily conserved from yeast to vertebrates and important for sister chromatid cohesion. We have addressed key points of the reviewer's

comments, modelling and confirming the interaction through additional biochemical experiments and through incorporation of the photoreactive nonnatural amino acid BPA, as summarized below.

Major points:

Figure 1.

According to the authors Scc2 (Figure 1a) is recruited by PCNA via a domain located in its C-terminal part. In vitro data showing that PCNA interacts in vitro with the PiP domain of Scc2 are in agreement with this model. Nevertheless, in order to strengthen this hypothesis, I suggest to the authors to test this interaction in vivo by performing for example Co-IP experiments between endogenous proteins (without addition of formaldehyde which could induce artifacts). If the authors do not succeed in revealing this interaction using classical coIP techniques, they could introduce the photoreactive amino acid Bpa in the pip domain of Scc2 in order to perform site-specific cross-linking assays (PMCID: PMC2483787). It would also be interesting to checking Chip-Seq if during the S phase PCNA colocalizes on the DNA with cohesin (and Scc2).

The data presented initially in Fig. 1e (now Fig. 2c) show in vitro interaction between PCNA and the Scc2 C-terminal peptide containing the PIP. Moreover, in Fig. 5a-d and Extended Data Fig. 8, we find similar interactions between vertebrate PCNA and the NIPBL Cterminus containing PIPs. We have now purified a nearly full-length Scc2 protein (only lacking the unstructured N-terminal 393 residues necessary for binding Scc4) and show that this GST-Scc2 fusion interacts directly with yeast PCNA in vitro (Fig. 2d and Extended Data Fig. 2b,c). We find that this binding depends largely on the PIP of Scc2 and is abolished in vitro by a C-terminal deletion of the last 168 residues of Scc2, suggesting that besides PIP other residues of Scc2 C terminus contribute to PCNA binding. We also modeled the interactions between PCNA and Scc2/NIPBL PIP using AlphaFold-Multimer in budding yeast (in comparison with Pol32 PIP, Fig. 2b), *S. pombe* and *G. gallus* (Extended Data Fig. 9). Collectively, these results indicate that Scc2 and NIPBL interact directly with PCNA via PIP(s) present in the C-terminus of the cohesin loader.

The interaction between budding yeast Scc2 and PCNA in vivo is not obvious by classical co-IP techniques and was detected using formaldehyde crosslinking (as presented initially in Extended Data Fig. 3b, c), but is observed in both human and avian cells (see initial Fig. 4a and Extended data Fig. 5a). The reviewer suggested that we introduce the non-natural photoreactive amino acid BPA in the PIP domain of Scc2 to perform site-specific crosslinking assays in vivo. As these assays require knowledge of the contact points between Scc2 PIP and PCNA, we modeled the interaction by AlphaFold-Multimer, and introduced BPA at 2 residues in Scc2, Q1475 and T1493 (new Extended Data Fig. 5d). We detected cross-links between Scc2 and PCNA in cells expressing C-terminally 3MYC-tagged Pol30 and Nterminally 3HA-tagged Scc2-Q1475BPA or Scc2-

T1493BPA, specifically following UV irradiation (new Extended Data Fig. 5e). Altogether, these results strongly support the interaction between PCNA and the cohesin loader Scc2/NIPBL via its C-terminal PIP motif.

Regarding ChIP-seq experiments between PCNA and the cohesin loader, they are of interest to examine in future works, but currently pose significant technical challenges due to the low levels of the cohesin loader interacting with chromatin and its rapid hopping between chromosomal cohesin rings after loading (Rhodes et al, *Elife*, 2017 and Petela et al, *Mol Cell*, 2018). What has been previously reported nevertheless (Tittel-Elmer et al, *Mol Cell*, 2012) is that in budding yeast, cohesin is recruited to active replication origins, as detected by Scc1 ChIP, and spreads along DNA as replication forks progress. However, coming close to the point of interest of the reviewer, we are showing that in vertebrate cells the amount of cohesin associating with nascent strands is reduced by truncation in the C-terminal tail of NIPBL containing PIPs and mediating the interaction with PCNA (see Fig. 5e).

Figure 3.

For clarity, I advise the authors to explain why it is relevant to use the *scc2-E822K* mutant to address the importance of the PIP domain of Scc2 in the recruitment of the loader to chromatin. The same is true for the inactivation of Scc4. In addition, I think it would be interesting and important to analyze the localization of the *scc2-pip* mutant in a genetic context lacking *scc2-E822K*. To address this, it will be preferable to use calibrated Chip-Seq technology rather than Chip q-PCR or chromatin fractionation assays that do not allow genome-wide assessment of *scc2* recruitment. It will also be interesting to analyze using Chip-Seq effect of the *scc2-pip* mutant on cohesin localization on chromatin. Since the pip domain is important for Scc2 recruitment, it is expected that the *scc2-pip* mutant will reduce the amount of cohesin on DNA throughout the genome.

Scc4 is critical to recruit the cohesin loader Scc2 to chromatin at centromeres via the binding of Scc4 to the DDK-phosphorylated kinetochore protein Ctf19 (Hinshaw et al, *Cell*, 2017). Moreover, Scc4 was recently shown to mediate Scc2 recruitment to chromatin via binding to chromatin remodeler RSC (Munoz et al, *Mol Cell*, 2019). To completely exclude any possible Scc4-mediated chromatin localization of Scc2, we needed to use the *scc2-E822K* mutant that bypasses the requirement of Scc4 for cell viability (Petela et al, *Mol Cell*, 2018). In addition, this allowed us to assess the contribution of both Scc4 and Scc2 PIP to the recruitment of the cohesin loader to chromatin and the Smc3 acetylation levels, which reflect the amount of cohesive cohesin in the cells, upon Scc4 depletion (Fig. 4c,d and Extended Data Fig. 7e). Recent work revealed that Scc2 has a key role in clamping DNA onto engaged SMC heads of cohesin, and that Scc2-E822K might

function by enhancing DNA binding within the clamped state (Collier et al, *Elife*, 2020 and Petela et al, *Elife*, 2021). Regarding the Scc2-E822K protein, Kim Nasmyth lab used calibrated ChIP-seq and showed that it has an increase in binding specifically around centromeres compared to Scc2 (Petela et al, *Mol Cell*, 2018), suggesting that Scc2-E822K extensively co-localizes with cohesin within an interval of 10 kb on either side of *CEN* loading sites. The authors of this work favor the notion that E822K enables Scc2 that had associated with cohesin at *CEN* loading sites to persist during its subsequent translocation. Scc2-E822K only modestly enhances cohesin association elsewhere in the genome, such as at replicating regions, where we expect to see changes due to PIP as assessed by ChIP-qPCR experiments (see initial Fig. 3). Based on the reviewer's suggestion, we are now explaining better the rationale of using Scc2-E822K in the revised manuscript.

Importantly, we observe the cohesion defects with mutations in the Scc2-PIP alone or when the PIPs of NIPBL are removed (see initial Fig. 1g and 4i), besides overall reduced levels of scc2-E822K-pip and acetylated Smc3 on chromatin, with these defects being rescued by the presence of a functional PIP of Pol32, but not when it is mutated (see initial Fig. 3b). The reduced levels of cohesin on nascent strands labeled with BrdU were also observed (see initial Fig. 4e). Altogether, the results support the notion that the PIPs on Scc2/NIPBL represent an evolutionarily conserved mechanism of recruiting the cohesin loader and subsequently cohesin that is then stabilized by acetylation to replicating regions.

Moreover, while we show in functional assays that the PIPs of Scc2 and NIPBL are important for cohesion (see initial Fig. 1g and 4i) and Smc3 acetylation (see initial Fig. 3b), the field currently lacks information whether and how the locations of the cohesin on chromatin relate to cohesion. Another point to consider is that ChIP-seq information of Smc3 acetylation is missing in the field and therefore which cohesin molecules are actually cohesive among entire chromatin-bound cohesin pool is not known. Furthermore, there are results showing that Eco1 and ESCO1/2-mediated Smc3 acetylation also affects genome organization (Dauban et al, *Mol Cell*, 2020; van Ruiten et al, *Nat Struct Mol Biol*, 2022). As these questions are very complex and not essential for the mechanism we are presenting here, we hope the reviewer understands that we cannot address them in this revision.

Minor points

I advise authors to include in the introduction key references that have previously suggested an interaction between DNA replication machinery and cohesin:

-Skibbens RV, Corson LB, Koshland D, Hieter P (1999) Ctf7p is essential for sister chromatid cohesion and links mitotic chromosome structure to the DNA replication machinery. *Genes Dev* 13: 307–319.

-Toth A, Ciosk R, Uhlmann F, Galova M, Schleiffer A, et al. (1999) Yeast Cohesin complex requires a conserved protein, Eco1p(Ctf7), to establish cohesion between sister chromatids during DNA replication. *Genes Dev* 13: 320–333.

-Mayer, M.L., Gygi, S.P., Aebersold, R., and Hieter, P. (2001). Identification of RFC(Ctf18p, Ctf8p, Dcc1p): an alternative RFC complex required for sister chromatid cohesion in *S. cerevisiae*. *Mol. Cell* 7, 959–970.

We introduced these references in the manuscript.

Reviewer #2:

Remarks to the Author:

This manuscript by Psakhye et al. describes a unique role of PCNA in recruiting the cohesin loader, Scc2, during de novo cohesin loading. Cohesion occurs via two mechanisms. The first entails establishment of cohesion during S-phase from cohesins that are already loaded prior to DNA replication, whereas the second requires cohesin loading during S-phase in a manner that depends on PCNA and specific clamp loading complexes. While genetic determinants of this second phase were known, mechanisms were not. The authors identified a PCNA interacting motif, PIP box, in Scc2 to be important for binding PCNA and sister chromatid cohesion. They characterized the functions of the Scc2 PIP box using elegant genetic experiments that reveal the essentiality of the PIP box in the absence of Scc4 and its role in chromatin recruitment. The authors also show the role of PCNA in recruiting Scc2/NIPBL is conserved in higher organisms.

This discovery provides important information towards understanding mechanisms of cohesion. The experiments are designed properly and are well executed. There are, however, a few comments that should be addressed prior to publication.

We thank the reviewer for acknowledging the important information that our manuscript brings to the field and for appreciating the elegant experiments and their execution. We further improved the manuscript in the spirit of the reviewer's comments as summarized below.

Major comments:

1) The authors have shown the interaction between Scc2-E822K and PCNA by IP (Extended data 3). But it is unclear when this interaction is observed. For example, is this interaction present only when *scc4* is absent. Does *scc2* WT also interact with PCNA? Since the *de novo* pathway is concurrent to DNA replication, a prediction is this association occurs in S phase. The authors should thoroughly investigate the cell cycle dependency of this interaction.

We have extended our analysis of the interaction between Scc2 and PCNA in several ways.

First, we have purified a nearly full-length recombinant Scc2 budding yeast protein, which we show to interact *in vitro* with budding yeast PCNA (Pol30). We find that this interaction depends largely on the Scc2 PIP and is abolished by a C-terminal deletion of the last 168 residues of Scc2 (Fig. 2d). Moreover, we uncover a similar interaction between NIPBL and PCNA *in vitro*, with this interaction being reduced by a C-terminal truncation of the domain containing PIPs (Extended Data Fig. 8e).

The interaction between budding yeast Scc2 and PCNA is not observed by classical co-IP techniques *in vivo* and requires formaldehyde crosslinking (as presented initially in Extended Data Fig. 3b, c), but is observed in both human and avian cells (see initial Fig. 4a and Extended data Fig. 5a). The first reviewer suggested that we introduce the non-natural photoreactive amino acid BPA in the PIP domain of Scc2 to perform site-specific crosslinking assays *in vivo*. As these assays require knowledge of the contact points between Scc2 PIP and PCNA, we modeled the interaction by AlphaFold-Multimer (Extended Data Fig. 5d) and introduced BPA at 2 residues in Scc2, Q1475 and T1493. We detected cross-links between Scc2 and PCNA in cells expressing C-terminally 3MYC-tagged Pol30 and Nterminally 3HA-tagged Scc2-Q1475BPA or Scc2-T1493BPA specifically following UV irradiation (Extended Data Fig. 5e).

Because co-IPs can be reproducibly performed in human cells, we attempted to synchronize human TK6 cells in G1, S and G2/M and study the levels of PCNA interacting with NIPBL in different cell cycle phases. We observe higher levels of PCNA interacting with NIPBL in cells enriched in S and G2/M compared to asynchronous populations and cells enriched in G1, a result included now in the Extended Data Fig. 8b. Moreover, we report reduced levels of the cohesin on nascent strands labeled with BrdU when NIPBL lacks its C-terminal PIPs (initial Fig. 4e), supporting the notion that the PIPs on Scc2/NIPBL represent an evolutionarily conserved mechanism of recruiting the cohesin loader to replicating regions.

Altogether, these results strongly support the notion that the cohesin loader Scc2/NIPBL is recruited to replicating strands via a mechanism involving its interaction with PCNA.

- 2) PCNA mutants Pol30-6 and pol30-79 could affect several other interactions (such as EcoI) contributing to benomyl sensitivity observed (Extended data 3). IPs detecting interaction differences between pol30-6 and pol30-79 and *scc2* would strengthen this data.

It is known that *pol30-6* and *pol30-79* contain mutations that cause disruptions of a surface cavity on the front face of the PCNA ring (Kondratieck et al, PLoS One, 2018) required to interact with several partners, likely by sequential binding and release of partners (Dovrat et al, PNAS, 2014). In this study we are showing that similar to Scc2 PIP, which becomes essential for viability in the absence of Scc4, mutations on the interface of PCNA engaged in sequential binding to partners becomes either essential (*pol30-6*) or important (*pol30-79*) for viability in the *scc4* null mutant kept alive by the *Scc2-E822K* mutant (Petela et al, 2008). Similarly, the disassembly-prone PCNA mutant *pol30-D150E* shows additive sensitivity to benomyl when combined with *scc4 scc2-E822K* double mutant (Extended Data Fig. 4e). Moreover, C-terminal tagging of PCNA with 3MYC, which we now show using alphaFold-Multimer prediction to affect the front face of the PCNA ring and to interfere with Scc2 PIP binding (Extended Data Fig. 6b), also exhibits additive sensitivity to benomyl when combined with *scc4 scc2-E822K* double mutant (Extended Data Fig. 6a). Based on different lines of experimentation, we infer that this is due to reduced levels of the cohesin loader on chromatin. Based on the genetic nature of these results, they are included as extended information.

- 3) In Fig. 3a the authors have shown chromatin association of the different Scc2 mutants and suggested reduced chromatin association of the pip box mutants. But in Fig. 3a total levels (WCE) of the pip mutants appears to be reduced. This suggests overall reduced stability/expression of these mutants rather than a defective chromatin recruitment. Does this affect interpretation of results?

We indeed observe reduced levels of Scc2-PIP proteins in WCE and chromatin fractions. Importantly, replacement of the PIP in Scc2 with the functional one of Pol32 rescues the defects in *scc2-PIP*, whereas point mutations in Pol32 PIP reactivate the defects. For these reasons, we think that the anchoring of the cohesin loader on PCNA stabilizes the Scc2 loader on chromatin. In this case, the recruitment of the loader on chromatin is also coupled with its stability. Furthermore, depletion of Scc4 that localizes Scc2 to chromatin by other means, through interaction with DDK-phosphorylated kinetochore protein Ctf19 (Hinshaw et al, Cell, 2017) and binding to chromatin remodeler RSC (Munoz et al, Mol Cell, 2019), also negatively affects Scc2 levels and shows additive effects when combined with the *scc2-pip* mutation (Extended Data Fig. 7e).

- 4) Is the contribution of Scc2 PIP and Scc4 in recruiting Scc2 cell cycle dependent?

This relates to the first question. We observe that Scc2 interacts directly with PCNA in vitro, in a manner independent of Scc2 binding to Scc4 (Fig. 2d). In human cells, the interaction between NIPBL and PCNA is increased in S and G2/M (Extended Data Fig. 8b), consistent with a replication-dependent recruitment mechanism of the cohesin loader.

5) Mutations of single pip boxes in NIPBL seem to affect the interactions with PCNA only partially. A triple mutant of all 3 pip boxes or IPs with NIPBL DC would address this issue more clearly.

Indeed, we have modeled the interactions and found that PIP2 and PIP3 are predicted to interact with PCNA by alphaFold-Multimer (Extended Data Fig. 9). Moreover, full-length NIPBL interacts with PCNA in vitro, with this interaction being weakened by a deletion of C-terminal domain containing PIPs (Extended Data Fig. 8e). Importantly, we report reduced levels of the cohesin on nascent strands labeled with BrdU when NIPBL lacks its C-terminal PIPs (initial Fig. 4e) due to the truncation of the last 195 residues.

Minor points

1) What is degrading NIPBL in Fig.4a?

We do not have the answer here. NIPBL is a very large protein and contains unstructured regions that may make it prone to degradation during immunoprecipitation. Using a new monoclonal antibody that recognizes 1-300 N-terminal residues, we were able to detect fulllength NIPBL after immunoprecipitation (Extended Data Fig. 8b).

Decision Letter, first revision:

Message: Our ref: NSMB-A46558A

24th Feb 2023

Dear Dr. Branzei,

Thank you for submitting your revised manuscript "PCNA recruits cohesin loader Scc2/NIPBL to ensure sister chromatid cohesion" (NSMB-A46558A). It has now been seen by the original referees and their comments are below. The reviewers find that the paper has improved in revision, and therefore we'll be happy in principle to publish it in Nature Structural & Molecular Biology, pending minor revisions to satisfy the referees' final requests and to comply with our editorial and formatting guidelines.

[EDITOR: REMOVE IF WORD DOCUMENT IS AVAILABLE]

To facilitate our work at this stage, we would appreciate if you could send us the main text as a word file. Please make sure to copy the NSMB account (cc'ed above).

Sincerely,

Carolina Perdigoto, PhD
Chief Editor
Nature Structural & Molecular Biology
orcid.org/0000-0002-5783-7106

Reviewer #1 (Remarks to the Author):

I would like to thank the authors for their articulate responses and the effort they made to address some of the reviewers' requests.

The authors have indeed made the effort to find residues of Scc2 that interact with POL30, I appreciate that. However, extended figure 5e is not convincing. The lower panel (anti-myc shows) shows two crosslink products. Which of these two products corresponds to the crosslink between Pol32 and Scc2? On the top panel (anti HA), it is difficult to see the crosslink product between Scc2 and Pol32. I encourage the authors to provide a better document.

I find interesting the fact that replacing Scc2 PIP by the Pol32 PIP suppresses the lethality associated with the absence of scc4. This experiment validates in some way that Scc2 has a PIP motif. Nevertheless, the importance of this result does not stand out because it is drowned in the middle of synthetic lethality results which are difficult to interpret and which drown the message of the paper. For example, in order to study the role of the PIP motif on cohesin loading, the authors decided to combine the scc2 pip mutant with the scc2 E822K mutation (Figure 3). The logic behind that is to exclude any Scc4-mediated cohesin loading. Their genetic analysis revealed that in scc2E822K scc4 genetic background Scc2 PIP motif becomes essential. According to the authors this result suggests that the Scc2 PIP motif is involved in the recruitment of Scc2 and cohesin to DNA. However, there are others possible interpretations for this result. One may envisage that the pip mutant prevents the scc2E822K mutation from suppressing the absence of Scc4. It is also possible that the combination of the PIP mutant with E822K changes the conformation or stability of Scc2. These mutation combinations do have an effect on the stability of Scc2, as shown in Figure 4a and b. I would also like to note that the effect on Scc2 stability is seen when scc2E822Kpip is expressed from an ADH1 promoter. This effect

on Scc2 stability may make the interpretation of the results difficult. Indeed, the effect of scc2E822K pip on Smc3 acetylation on cohesin and Scc2 loading could be due to the fact that there is less Scc2 in this cellular context and does not reveal any role of Scc2 PIP domain in cohesin loading during replication.

I again recommend that the authors to evaluate the effect of scc2 pip mutant on cohesin loading (during replication) as it is difficult to interpret the data obtained from scc2 E822K pip. Whatever the result, it will be interesting to know.

Reviewer #2 (Remarks to the Author):

The authors have satisfactorily addressed all of my concerns. This stands as an important study to understand replication associated cohesion.

Author Rebuttal, first revision:

Reviewers' Comments:

Reviewer #1:

Remarks to the Author:

I would like to thank the authors for their articulate responses and the effort they made to address some of the reviewers' requests.

The authors have indeed made the effort to find residues of Scc2 that interact with POL30, I appreciate that. However, extended figure 5e is not convincing. The lower panel (anti-myc shows) shows two crosslink products. Which of these two products corresponds to the crosslink between Pol32 and Scc2? On the top panel (anti HA), it is difficult to see the crosslink product between Scc2 and Pol32. I encourage the authors to provide a better document.

We are happy that the reviewer appreciates the efforts to further strengthen our finding that cohesin loader Scc2/NIPBL interacts with PCNA. Now, besides demonstrating the direct interaction between PCNA (yeast Pol30) and the C-terminal PIP of Scc2 fused to GST in vitro (Figure 2c), we show that nearly full-length Scc2 fused to GST (GST-scc2C1100), which only lacks its N-terminal part necessary for binding to Scc4, interacts with PCNA (Figure 2d and Extended Data Figure 2b). Importantly, the binding to PCNA in vitro largely depends on the PIP of Scc2 and fully relies on the residues 1326-1493 of its C-terminus. Moreover, we used AlphaFold2 and AlphaFold-Multimer to predict the structure of Scc2 PIP, not resolved previously in the cryo-EM structure of cohesin with the loader (PDB 6ZZ6) due to its flexibility, and its interaction with PCNA (Figure

2b and Extended Data Figure 2c). Furthermore, we previously demonstrated binding of human and chicken NIPBL to PCNA in vivo by performing co-IP experiments (Figure 5a and Extended Data Figure 8a), and showed that the C-terminus of NIPBL containing PIPs fused to GST interacts with PCNA in vitro (Figure 5d). As conventional co-IP experiments in yeast were not successful in detecting interaction between Pol30 and Scc2, we previously used in vivo formaldehyde cross-linking to capture the interaction followed by Ni-NTA pull-down of N-terminally His-tagged Scc2 under fully denaturing conditions in order to demonstrate its binding to PCNA (Extended Data Figure 5b). Now, following reviewer's suggestion we introduced the unnatural photoreactive amino acid BPA at 2 positions flanking C-terminal PIP of Scc2 (Extended Data Figure 5d) and demonstrated cross-linking of Scc2 to Pol30 in vivo, specifically following UV-irradiation of the cells (Extended Data Figure 5e). As cohesin loader is a very low abundant protein, we had to perform IP in order to enrich it along with its cross-linked species. Despite these obstacles and susceptibility of Scc2 to undergo proteolysis during the immunoprecipitation we were able to detect up-shifted cross-linked species for both Scc2 and PCNA (yeast Pol30). We believe that all these interaction studies both in vitro and in vivo from different species (yeast, chicken and human cells) support the finding that cohesin loader Scc2/NIPBL interacts with PCNA. Consistently, AlphaFold-Multimer predictions further support that the interaction between PCNA and the C-terminal PIPs of cohesin loaders is conserved from yeast to vertebrates (Extended Data Figure 9).

I find interesting the fact that replacing Scc2 PIP by the Pol32 PIP suppresses the lethality associated with the absence of *scc4*. This experiment validates in some way that Scc2 has a PIP motif. Nevertheless, the importance of this result does not stand out because it is drowned in the middle of synthetic lethality results which are difficult to interpret and which drown the message of the paper. For example, in order to study the role of the PIP motif on cohesin loading, the authors decided to combine the *scc2 pip* mutant with the *scc2 E822K* mutation (Figure 3). The logic behind that is to exclude any Scc4-mediated cohesin loading. Their genetic analysis revealed that in *scc2E822K scc4* genetic background Scc2 PIP motif becomes essential. According to the authors this result suggests that the Scc2 PIP motif is involved in the recruitment of Scc2 and cohesin to DNA. However, there are others possible interpretations for this result. One may envisage that the *pip* mutant prevents the *scc2E822K* mutation from suppressing the absence of Scc4. It is also possible that the combination of the PIP mutant with E822K changes the conformation or stability of Scc2. These mutation combinations do have an effect on the stability of Scc2, as shown in Figure 4a and b. I would also like to note that the effect on Scc2 stability is seen when *scc2E822Kpip* is expressed from an ADH1 promoter. This effect on Scc2 stability may make the interpretation of the results difficult. Indeed, the effect of *scc2E822K pip* on Smc3 acetylation on cohesin and Scc2 loading could be due to the fact that there is less Scc2 in this cellular context and does not reveal any role of Scc2 PIP domain in cohesin loading during replication.

We would like to emphasize that *scc2-pip* shows cohesion defects, further exacerbated when components of the cohesin conversion pathway are mutated (Figure 2 and Extended Data Figure 3e-h), without E822K mutation. We utilized *scc2-E822K* mutation in order to use *scc4* delta

background thus allowing us to exclude any Scc4-mediated chromatin recruitment of Scc2 and expose the importance of the PIP. Importantly, the level of Scc2 is not affected by mutating PIP (Extended Data Figure 3a). We like to note that we worked also in conditions in which *scc2-E822K-pip* is expressed from the strong *pADH1* promoter and has similar levels with *scc2-E822K* (both about 100 fold higher than *scc2* variants expressed from the endogenous *SCC2* promoter) (Figure 4a). Also under these conditions, we detect a decrease in the levels of the Scc2-E822K recruitment to chromatin in a manner dependent on its PIP (Figure 4c, 4d). Moreover, loss of PIPs in the C-terminus of NIPBL that only reduces the interaction with PCNA in vitro (Extended Data Fig. 8e) causes increase in chromosome loss (Figure 6b) and cohesion defects in parallel with the Chl1/DDX11 pathway (Figure 6d).

I again recommend that the authors to evaluate the effect of *scc2 pip* mutant on cohesin loading (during replication) as it is difficult to interpret the data obtained from *scc2 E822K pip*. Whatever the result, it will be interesting to know.

The new paper from Frank Uhlmann's lab in Cell (Minamino et al, 2023), provides more evidence of why we would not expect to observe a detectable defect of the *scc2-pip* mutant in loading cohesin while still causing a cohesion defect as we detect in our paper (Figure 2f). The recent study from Uhlmann lab is reconstituting cohesin acetylation by Eco1 in vitro. Specifically, in Figure 7B of their work, they show that both WT Scc2-Scc4 loader complex (Scc2-4) and C-terminal fragment of Scc2 that does not longer bind Scc4, called Scc2C, are proficient in loading cohesin (Smc1-PK) on DNA, but Scc2C is defective in cohesin acetylation (ac-Smc3). Importantly, however, Scc2C is tagged C-terminally with HA-tag, whereas Scc2 within Scc2-4 does not have C-terminal HA tag (see Figure S7D), therefore making the comparison of Scc2-4 with Scc2C inappropriate. Based on our findings, we can conclude that this Scc2C protein bearing HA-tag adjacent to PIP is defective in the interaction with PCNA, as we show that tagging Scc2 C-terminally impairs the functionality of the PIP (our study Figure 3f and Extended Data Figure 3a-d). Moreover, we show in vitro that Scc2 alone without Scc4 interacts with PCNA, in a PIP dependent manner (our study Figure 2d and Extended Data Figure 2b). Thus, inability of Scc2 to interact with PCNA via its C-terminal PIP results in defective Smc3 acetylation that marks cohesive cohesin, while overall cohesin DNA loading (Smc1-PK) is not being affected by the loss of Scc2 binding to PCNA (Minamino et al, 2023, Figure 7B). It is most likely that *scc2-pip* is affecting the loading of a small pool of cohesin, proximal to PCNA, ultimately leading to a defect in cohesin acetylation. Please note that there are multiple pathways contributing to cohesin loading, including the one mediated by Chl1 and multiple ones mediated by Scc4, making the contribution of this newly identified module related to Scc2 recruitment via PCNA, difficult to detect technically in terms of overall cohesin amounts on DNA/chromatin.

Reviewer #2:

Remarks to the Author:

The authors have satisfactorily addressed all of my concerns. This stands as an important study to understand replication associated cohesion.

We are happy that the reviewer finds all concerns addressed, and acknowledges our findings as being important for understanding how sister chromatid cohesion is being generated during replication.

Final Decision Letter:

Message 12th Jul 2023

:

Dear Dr. Branzei,

Please accept my sincere apologies in the delay in sending the final decision on your study, I am afraid I have been on medical leave for the past weeks.

We are now happy to accept your revised paper "PCNA recruits cohesin loader Scc2/NIPBL to ensure sister chromatid cohesion" for publication as a Article in Nature Structural & Molecular Biology.

As soon as your article is published, you can generate your shareable link by entering the DOI of your article here: http://authors.springernature.com/share. Corresponding authors will also receive an automated email with the shareable link

Your paper will be published online soon after we receive proof corrections and will appear in print in the next available issue. You can find out your date of online publication by contacting the production team shortly after sending your proof corrections. Content is published online weekly on Mondays and Thursdays, and the embargo is set at 16:00 London time (GMT)/11:00 am US Eastern time (EST) on the day of publication. Now is the time to inform your Public Relations or Press Office about your paper, as they might be interested in promoting its publication. This will allow them time to prepare an accurate and satisfactory press release. Include your manuscript tracking number (NSMB-A46558B) and our journal name, which they will need when they contact our press office.

About one week before your paper is published online, we shall be distributing a press release to news organizations worldwide, which may very well include details of your work. We are happy for your institution or funding agency to prepare its own press release, but it must mention the embargo date and Nature Structural & Molecular Biology. If you or your Press Office have any enquiries in the meantime, please contact press@nature.com.

An online order form for reprints of your paper is available at https://www.nature.com/reprints/author-reprints.html. Please let your coauthors and your institutions' public affairs office know that they are also welcome to order reprints by this method.

Please note that *Nature Structural & Molecular Biology* is a Transformative Journal (TJ). Authors may publish their research with us through the traditional subscription access route or make their paper immediately open access through payment of an article-processing charge (APC). Authors will not be required to make a final decision about access to their article until it has been accepted. [Find out more about Transformative Journals](https://www.springernature.com/gp/open-research/transformative-journals)

Authors may need to take specific actions to achieve [compliance with funder and institutional open access mandates](https://www.springernature.com/gp/open-research/funding/policy-compliance-faqs). If your research is supported by a funder that requires immediate open access (e.g. according to [Plan S principles](https://www.springernature.com/gp/open-research/plan-s-compliance)) then you should select the gold OA route, and we will direct you to the compliant route where possible. For authors selecting the subscription publication route, the journal's standard licensing terms will need to be accepted, including [self-archiving policies](https://www.springernature.com/gp/open-research/policies/journal-policies). Those licensing terms will supersede any other terms that the author or any third party may assert apply to any version of the manuscript.

Sincerely,

Carolina Perdigoto, PhD
Chief Editor
Nature Structural & Molecular Biology
orcid.org/0000-0002-5783-7106
